# Wnt induces FZD5/8 endocytosis and degradation and the involvement of RSPO-ZNRF3/RNF43 and DVL

**Dong Luo[1†], Jing Zheng[1†], Shuning Lv[1†], Ren Sheng[2], Maorong Chen[3], Xi He[4], Xinjun Zhang[1,5]\***

[1]Key Laboratory of Molecular Biophysics of the Ministry of Education, College of Life Science and Technology, Huazhong University of Science and Technology, Wuhan, China; Genetic Diseases Key Laboratory of Sichuan Province and the Department of Laboratory Medicine, Sichuan Provincial People's Hospital, University of Electronic Science and Technology of China, Chengdu, China; [2]College of Life and Health Science, Northeastern University, Shenyang, China; [3]Center for Life Sciences, School of Life Sciences, Yunnan University, Kunming, China; [4]The F. M. Kirby Neurobiology Center, Boston Children's Hospital, Department of Neurology, Harvard Medical School, Boston, United States; [5]Research Unit for Blindness Prevention of the Chinese Academy of Medical Sciences (2019RU026), Sichuan Academy of Medical Sciences and Sichuan Provincial People's Hospital, Chengdu, China

**\*For correspondence:**
xzhang@uestc.edu.cn

[†]These authors contributed equally to this work

## eLife Assessment

This study presents **important** findings demonstrating that the internalization and degradation of FZD5 and FZD8, two of the ten Frizzled proteins, are WNT dependent and do not involve DVL. The evidence supporting the claims of the authors is **convincing**. This research will be of interest to biologists specializing in Wnt signaling, cancer, and regenerative medicine.

**Abstract** Frizzled (FZD) proteins are the principal receptors of the Wnt signaling pathway. However, whether Wnt ligands induce FZD endocytosis and degradation remains elusive. The transmembrane E3 ubiquitin ligases ZNRF3 and RNF43 promote the endocytosis and degradation of FZD receptors to inhibit Wnt signaling, and their function is antagonized by R-spondin (RSPO) proteins. However, the dependency of RSPO-ZNRF3/RNF43-mediated FZD endocytosis and degradation on Wnt stimulation, as well as the specificity of this degradation for different FZD, remains unclear. Here, we demonstrated that Wnt specifically induces FZD5/8 endocytosis and degradation in a ZNRF3/RNF43-dependent manner. ZNRF3/RNF43 selectively targets FZD5/8 for degradation upon Wnt stimulation. RSPO1 enhances Wnt signaling by specifically stabilizing FZD5/8. Wnt promotes the interaction between FZD5 and RNF43. We further demonstrated that DVL proteins promote ligand-independent endocytosis of FZD but are dispensable for Wnt-induced FZD5/8 endocytosis and degradation. Our results reveal a novel negative regulatory mechanism of Wnt signaling at the receptor level and illuminate the mechanism by which RSPO-ZNRF3/RNF43 regulates Wnt signaling in human cells, which may provide new insights into regenerative medicine and cancer therapy.

## Introduction

The evolutionarily conserved Wnt signaling pathways play pivotal roles in embryonic development and adult tissue homeostasis (*Clevers and Nusse, 2012*; *Nusse and Clevers, 2017*; *Rim et al., 2022*; *Stapornwongkul and Vincent, 2021*; *Steinhart and Angers, 2018*). There are nineteen Wnt ligands and ten Frizzled (FZD) receptors in mammals (*Wang et al., 2016*). Wnt-FZD complexes activate several intracellular signaling pathways, including the canonical (Wnt/β-catenin) and noncanonical pathways. In the Wnt/β-catenin pathway, certain Wnt-FZD complexes engage and phosphorylate the coreceptors LRP5/6, inhibiting β-catenin phosphorylation and degradation, and stabilized cytosolic β-catenin is translocated to the nucleus, where it activates the TCF/LEF1 transcription factors to induce target gene expression. The noncanonical pathways, mainly the Wnt/PCP and Wnt/Ca$^{2+}$ pathways, are activated by Wnt-FZD complexes with the assistance of the coreceptors ROR1/2, VANGL, RYK, and/or PTK7. These pathways regulate cell and tissue polarity, movement, and gene expression (*Niehrs, 2012*). Wnt signaling pathways are tightly regulated at different levels to maintain normal physiological functions (*Jiang and Cong, 2016*; *Rim et al., 2022*). Multiple Wnt pathway mutations have been identified as driver events in human cancers, and several therapeutic strategies have been developed (*Anastas and Moon, 2013*; *Bugter et al., 2021*; *O'Brien et al., 2023*).

Receptor endocytosis serves as a common regulatory mechanism across multiple signaling pathways. Numerous studies have reported that receptor endocytosis positively influences both the canonical and noncanonical Wnt pathways by facilitating or amplifying intracellular signaling (*Chen et al., 2003*; *Yamamoto et al., 2006*; *Yu et al., 2007*; *Zhang et al., 2017*). Conversely, ligand-induced receptor endocytosis and degradation play critical roles in controlling signal strength and duration and are typical negative regulatory mechanisms in receptor tyrosine kinase pathways (*Alexander, 1998*; *Haglund and Dikic, 2012*; *Marmor and Yarden, 2004*; *Sorkin and Waters, 1993*). However, whether Wnt induces FZD receptor endocytosis and degradation remains poorly characterized.

ZNRF3 and RNF43 are closely related transmembrane E3 ubiquitin ligases that are induced by Wnt/β-catenin signaling and antagonize Wnt signaling by promoting FZD receptor endocytosis and degradation (*Bugter et al., 2024*; *Hao et al., 2012*; *Koo et al., 2012*). However, how ZNRF3/RNF43 recognize FZD receptors is still controversial (*Farnhammer et al., 2023*; *Hao et al., 2016*; *Tsukiyama et al., 2021*). A recent report suggested that ZNRF3/RNF43 ubiquitinates FZD receptors through binding to DVL proteins, which were previously thought to regulate FZD protein endocytosis (*Chen et al., 2003*; *Jiang et al., 2015*). Another study proposed that ZNRF3/RNF43 interact with FZD receptors via their extracellular protease-associated (PA) domain (*Tsukiyama et al., 2015*). R-spondin (RSPO) family proteins of stem cell growth factors were reported to potentiate Wnt signaling by binding to and degrading ZNRF3/RNF43 with the help of LGR4 or LGR5 (*Hao et al., 2012*; *Kazanskaya et al., 2004*; *Park et al., 2020*; *Xie et al., 2013*). However, recent studies have indicated that RSPO can activate Wnt signaling through ZNRF3/RNF43 independently of LGRs (*Dubey et al., 2020*; *Lebensohn and Rohatgi, 2018*; *Szenker-Ravi et al., 2018*). Whether RSPO-ZNRF3/RNF43-mediated regulation of FZD receptors relies on Wnt stimulation and whether there are specificities toward different FZDs remains unclear.

In this study, we first demonstrated that Wnt specifically induces FZD5/8 endocytosis and degradation in a ZNRF3/RNF43-dependent manner. We further demonstrated that DVL proteins promote ligand-independent FZD receptor endocytosis but do not affect Wnt-induced FZD5/8-specific endocytosis and degradation. Furthermore, we reevaluated the functions of ZNRF3/RNF43 and RSPO1 in regulating Wnt signaling. Our results suggest that RSPO1-ZNRF3/RNF43 specifically regulates FZD5/8 levels in the presence of Wnt ligands. We also showed that Wnt induces the FZD5–RNF43 interaction at the plasma membrane. Our study reveals a novel mechanism for the regulation of Wnt signaling at the receptor level and provides further insights into the function of RSPO1-ZNRF3/RNF43 in the Wnt signaling pathway.

## Results

### Wnt specifically induces FZD5/8 endocytosis and degradation

To investigate whether Wnt induces endocytosis and degradation of FZD receptors, HEK293A cells stably expressing each of the ten V5-FZDs (with the V5 tag added to the amino terminus of each FZD after the signal peptide) were treated with control, Wnt3a-conditioned or Wnt5a-conditioned

medium (CM), and the FZD levels on the cell surface were examined via flow cytometry with an anti-V5 antibody. Both Wnt3a and Wnt5a dramatically reduced the expression of FZD5 and FZD8 on the cell surface without significantly affecting other FZD receptors (*Figure 1A*). Immunoblotting revealed that both Wnt3a and Wnt5a reduced the mature protein levels of both FZD5 and FZD8 (the upper bands of FZD5 and FZD8) (*Figure 1B*; *Chen et al., 2020*; *Koo et al., 2012*). Treatment with the lysosomal inhibitor bafilomycin A1 rescued the mature form of FZD5 that was reduced by Wnt3a and Wnt5a (*Figure 1C*). Huh7 cells stably expressing each of ten V5-FZDs were also treated with control, Wnt3a, or Wnt5a CM, and both Wnt3a and Wnt5a specifically reduced FZD5 and FZD8 on the cell surface (*Figure 1—figure supplement 1A*). U2OS cells stably expressing V5-FZD1/FZD4/FZD5/FZD7 were treated with control, Wnt3a, Wnt5a CM, and only the mature form of FZD5 showed decreased levels upon Wnt3a or Wnt5a treatment (*Figure 1—figure supplement 1B-D*). These results suggest that Wnt3a and Wnt5a specifically promote the endocytosis and lysosomal degradation of FZD5 and FZD8.

FZD receptors are classified into four clusters, with sequence variations predominantly in the extra-cellular CRD domain and C-terminal tail (*Huang and Klein, 2004*). To investigate which domain is required for Wnt-induced FZD5 endocytosis, we generated FZD5 constructs with deletion of the CRD domain (△CRD) or C-terminal tail (△C) and analyzed them via flow cytometry. The results indicated that the CRD domain, but not the C-terminal tail, is crucial for Wnt3a/Wnt5a-induced FZD5 endocytosis (*Figure 1D–F*). Additionally, we constructed chimeric FZD proteins with swapped CRD domains, as illustrated in the schematic diagram (*Figure 1G*). For example, FZD4CRD-FZD5 consists of the CRD domain of FZD4 and FZD5 lacking the CRD domain. Wnt3a or Wnt5a treatment reduced the cell surface levels of FZD5CRD-FZD4 and FZD5CRD-FZD7 but had little effect on the cell surface levels of FZD4CRD-FZD5 or FZD7CRD-FZD5 (*Figure 1H*). These results suggest that Wnt-induced FZD5/8-specific endocytosis relies on their CRD domains.

As the commercially available anti-FZD antibodies either cannot detect endogenous proteins or can only recognize the immature form (data not shown), to visualize endogenous FZD5, we generated an FZD5 knock-in HEK293A cell line in which a V5 tag was inserted into the carboxyl terminus of the FZD5 locus before the stop codon via the CRISPR/Cas9 system and homologous recombination. Like V5-FZD5 overexpression, the endogenous FZD5-V5 protein exhibited two major bands on immunoblot-ting, and both the Wnt3a and Wnt5a treatments reduced the intensity of the upper band representing the mature form of FZD5, which occurred as early as one hour after treatment (*Figure 2A*). Using the same approach, we generated a FZD7-V5 knock-in cell line and observed that neither Wnt3a nor Wnt5a induced degradation of endogenous FZD7 (*Figure 2B*). Furthermore, treatment with the Porcupine inhib-itor IWP-2 increased the level of the endogenous mature FZD5 protein but did not affect FZD7 levels (*Figure 2C*). Flow cytometry with anti-FZD5/8 or anti-FZD4 monoclonal antibodies revealed that IWP-2 treatment increased the cell surface levels of FZD5/8 but had little effect on FZD4 levels in HEK293A cells (*Figure 2D and E*; *Pan et al., 2021*; *Sachdev Sidhu et al., 2021*; *Steinhart et al., 2017*). IWP-2 treatment also increased the cell surface levels of FZD5/8 in Huh7, MCF7, and 769 P cells (*Figure 2—figure supple-ment 1A-C*). The specificity of the anti-FZD5/8 and anti-FZD4 antibodies was confirmed in HEK293A cells overexpressing all ten FZD receptors (*Figure 2—figure supplement 2*). These results further indicate that Wnt specifically induces the endocytosis and degradation of endogenous FZD5/8.

## Wnt-induced FZD5/8 degradation relies on ZNRF3/RNF43

ZNRF3 and RNF43 are homologous transmembrane E3 ubiquitin ligases that have been reported to target FZD receptors for endocytosis and degradation (*Farnhammer et al., 2023*; *Hao et al., 2012*; *Koo et al., 2012*). We wondered whether Wnt-induced FZD5/8 endocytosis and degradation depend on ZNRF3/RNF43. To address this, we generated a *ZNRF3/RNF43* double knockout HEK293A cell line (ZRDKO) via the CRISPR/Cas9 technique, which was verified by sequencing (*Figure 2—figure supple-ment 3*). Flow cytometry analysis with an anti-FZD5/8 antibody revealed that *ZNRF3/RNF43* double knockout increased FZD5/8 levels on the cell surface (*Figure 2F*), which is consistent with previous reports (*Hao et al., 2012*; *Koo et al., 2012*). In contrast, the membrane levels of FZD4 remained unchanged in ZRDKO cells (*Figure 2G*). We subsequently established an FZD5-V5 knock-in cell line based on ZRDKO. Immunoblotting revealed that neither Wnt3a nor Wnt5a induced the degrada-tion of endogenous mature FZD5 in ZRDKO cells (*Figure 2H*). Additionally, IWP-2 treatment did not further increase mature FZD5 levels in ZRDKO cells (*Figure 2I*).

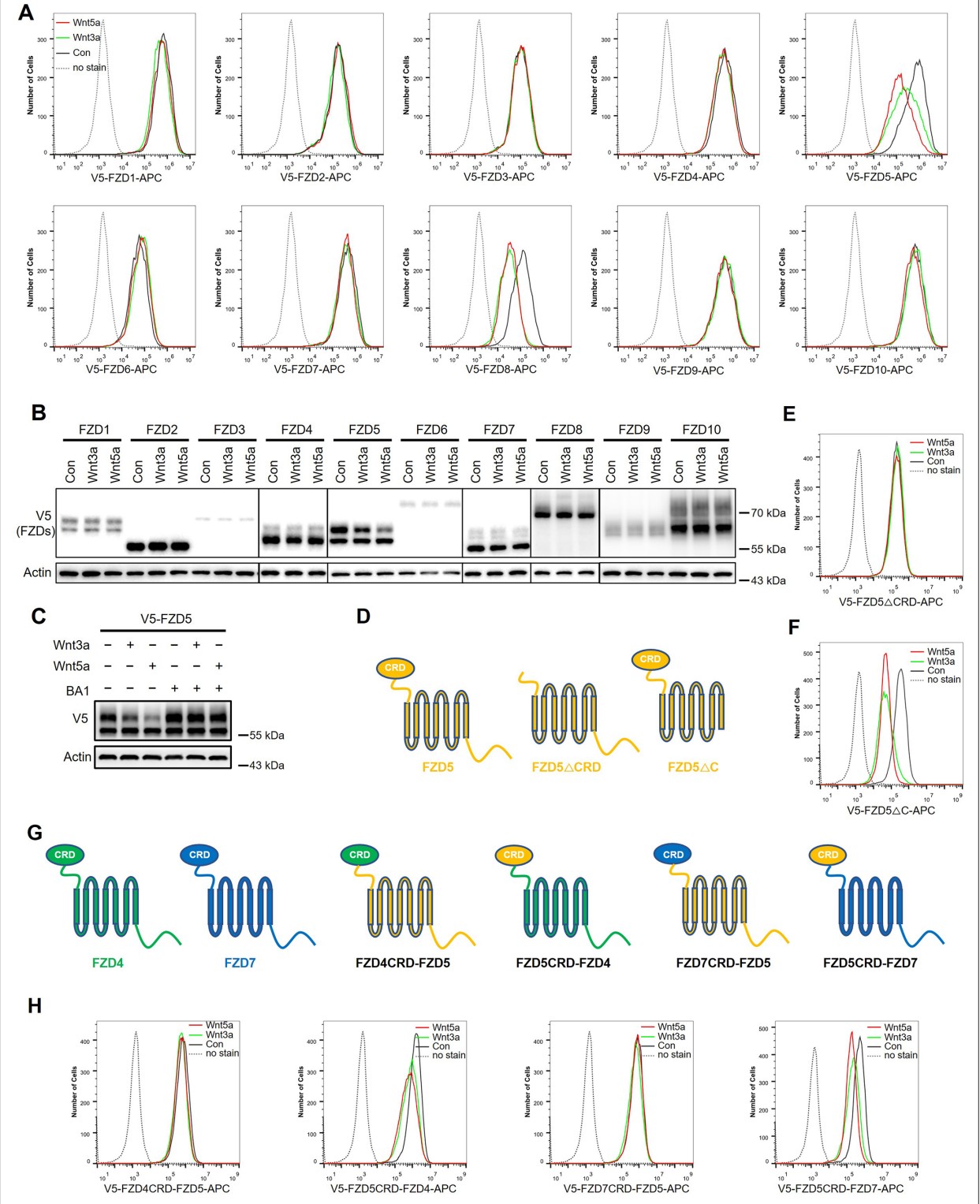

**Figure 1.** Wnt induces FZD5/8 endocytosis and degradation. (**A**) Wnt3a or Wnt5a specifically reduced the cell surface levels of V5-FZD5/8. HEK293A cells stably expressing each of the 10 V5-FZDs (FZD1 to 10) were treated with control, Wnt3a, or Wnt5a CM for 4 hours, and the cell surface levels of V5-FZDs were analyzed via flow cytometry using an anti-V5 antibody. (**B**) Wnt3a or Wnt5a specifically decreased the levels of mature forms of V5-FZD5/8. The whole cell lysates (WCLs) from the indicated cells treated as described in (**A**) were analyzed by immunoblotting with the indicated antibodies. (**C**) Bafilomycin A1 (BA1) restored Wnt3a or Wnt5a induced FZD5 degradation. (**D**) Schematic diagram of FZD5 constructs: full-length FZD5, FZD5 lacking the extracellular cysteine-rich domain (FZD5△CRD), and FZD5 lacking the intracellular C-terminus (FZD5△C). (**E**, **F**) HEK293A cells stably

*Figure 1 continued on next page*

Figure 1 continued

expressing the FZD5 truncation constructs shown in (**D**) were treated and analyzed as described in (**A**). (**G**) Schematic diagram of CRD-swapped chimeric FZDs. (**H**) HEK293A cells stably expressing the indicated chimeric FZD constructs shown in (**F**) were treated and analyzed as described in (**A**).

The online version of this article includes the following source data and figure supplement(s) for figure 1:

**Source data 1.** Raw unedited blots for *Figure 1*.

**Source data 2.** Uncropped and labeled blots for *Figure 1*.

**Figure supplement 1—source data 1.** Raw unedited blots for *Figure 1—figure supplement 1*.

**Figure supplement 1—source data 2.** Uncropped and labeled blots for *Figure 1—figure supplement 1*.

**Figure supplement 1.** Wnt induces FZD5/8 endocytosis and degradation.

RSPO proteins antagonize ZNRF3/RNF43, thereby potentiating Wnt signaling (*Hao et al., 2012*; *Xie et al., 2013*; *Yan et al., 2017*). Given that ZNRF3/RNF43 specifically degrades FZD5/8 in the presence of Wnt, we examined whether RSPO1 treatment could specifically increase FZD5/8 levels. Immunoblotting revealed that RSPO1 significantly increased the level of endogenous mature FZD5 but did not affect the FZD7 level (*Figure 2J*). Moreover, RSPO1 had little effect on FZD5 levels in ZRDKO cells (*Figure 2K*). To further detect the precise changes of endogenous FZD5, we introduced surface biotinylation assay in HEK293A FZD5KI cells. Results showed that Wnt3a and Wnt5a induced membrane FZD5 degradation, which was blocked by cotreatment with RSPO1 (*Figure 2L*). Flow cytometry analysis of wild-type (WT) HEK293A cells treated with IWP-2 or RSPO1 revealed that both treatments similarly elevated FZD5/8 levels on the cell surface (*Figure 2M*). However, RSPO1 treatment did not increase membrane FZD5/8 levels in ZRDKO cells or FZD4 levels in WT cells (*Figure 2N and O*).

Taken together, these results suggest that ZNRF3/RNF43 specifically degrades FZD5/8 under Wnt stimulation, which is antagonized by RSPO.

## DVL proteins regulate Wnt-independent FZD endocytosis and are dispensable for Wnt-induced FZD5/8 endocytosis or degradation

DVL proteins have been reported to regulate FZD endocytosis and degradation by binding to ZNRF3/RNF43 and recruiting it to FZD receptors (*Jiang et al., 2015*; *Yu et al., 2007*). We investigated whether DVL proteins are required for Wnt-induced FZD5/8 endocytosis and degradation. To address this, we generated a *DVL1/2/3* triple knockout HEK293A cell line (DVLTKO), which was verified by genomic sequencing and immunoblotting (*Figure 3A*, *Figure 3—figure supplement 1*). Flow cytometry analysis revealed that triple knockout of *DVL1/2/3* increased the cell surface levels of FZD5/8 (*Figure 3B*), suggesting that DVL proteins promote FZD5/8 endocytosis, which is consistent with a previous report (*Jiang et al., 2015*). However, both IWP-2 and RSPO1 treatments further elevated the cell surface levels of FZD5/8 in DVLTKO cells (*Figure 3C and D*), indicating that Wnt can still induce ZNRF3/RNF43-dependent FZD5/8 endocytosis even in the absence of DVL proteins. Furthermore, Wnt3a and Wnt5a reduced both the cell surface level and the mature form of the FZD5 protein in DVLTKO cells stably expressing V5-FZD5, as assessed by flow cytometry and immunoblotting (*Figure 3E and F*).

To visualize FZD endocytosis dynamics, we performed an antibody-labeled endocytosis assay. WT, DVLTKO, and DVLTKO re-expressing DVL2 (DVLTKO + DVL2) cells stably expressing V5-linker-FZD5 were first treated with IWP-2 overnight to inhibit the secretion of endogenous Wnt proteins. The cells were labeled with an anti-V5 antibody and then treated with control, Wnt3a, or Wnt5a CM for 1 hr (*Figure 3G*), and the internalized antibody was visualized by immunostaining. The linker sequence was used to separate the V5 tag from the FZD5 protein to avoid interference with Wnt3a or Wnt5a binding. Compared with WT cells, DVLTKO cells presented fewer FZD5 puncta in the control CM-treated group, and re-expressing DVL2 in DVLTKO cells elevated FZD5 puncta, whereas Wnt3a and Wnt5a increased the number of FZD5 puncta in both WT, DVLTKO, and DVLTKO + DVL2 cells, as quantified by the number of FZD5 puncta (*Figure 3H*). Compared with WT cells, DVLTKO cells presented decreased FZD7 internalization in control CM, which could also be rescued by re-expressing DVL2, and Wnt3a or Wnt5a CM had no effect on FZD7 endocytosis in either WT, DVLTKO, or DVLTKO + DVL2 cells (*Figure 3I and J*). The whole cell lysates from WT, DVLTKO, and DVLTKO + DVL2 cells expressing V5-linker-FZD5 or V5-linker-FZD7 were analyzed by immunoblotting (*Figure 3K*).

Collectively, these results suggest that DVL proteins participate in Wnt-independent FZD receptor endocytosis but are dispensable for Wnt-induced FZD5/8 endocytosis and degradation.

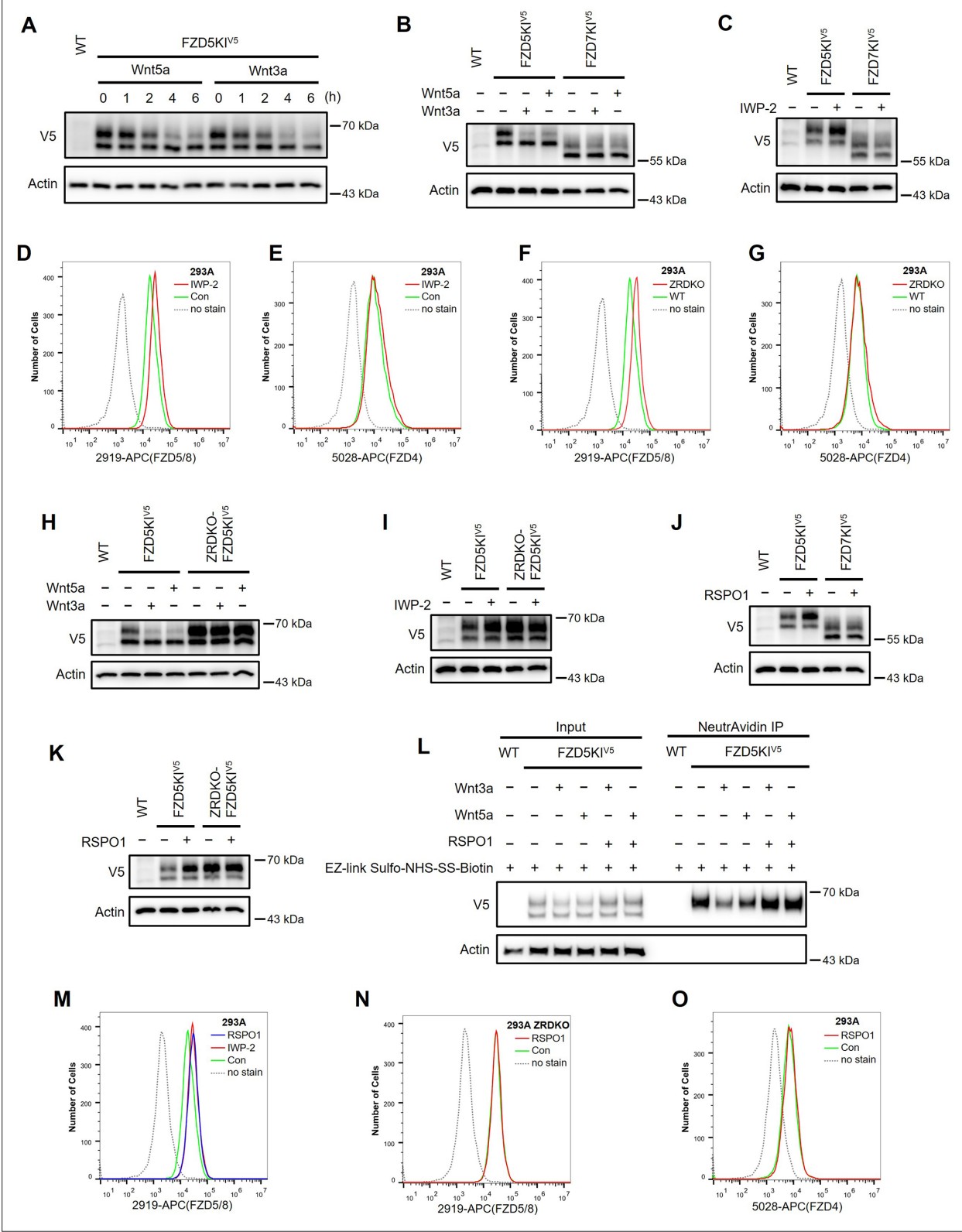

**Figure 2.** Wnt induces ZNRF3/RNF43-dependent FZD5/8 endocytosis and degradation. (**A**) Wnt3a or Wnt5a induced the degradation of endogenous FZD5. HEK293A cells with a V5 epitope tag knocked in at the C-terminus of endogenous FZD5 (FZD5KI) were treated with Wnt3a or Wnt5a CM for the indicated durations, and the WCLs were analyzed by immunoblotting with the indicated antibodies. Wild-type (WT) cells served as a negative control to confirm the specificity of the FZD5KI bands. The upper bands represent mature FZD5, whereas the lower bands represent immature FZD5.

*Figure 2 continued on next page*

*Figure 2 continued*

(**B**) Wnt3a or Wnt5a induced the degradation of endogenous FZD5 but not FZD7. FZD5KI or FZD7KI (generated as described for FZD5KI) HEK293A cells were treated with control, Wnt3a CM, or Wnt5a CM for 2 hr, and the WCLs were subjected to immunoblotting with the indicated antibodies. (**C**) The Porcupine inhibitor IWP-2 increased the level of the endogenous mature form of FZD5 but not FZD7. FZD5KI or FZD7KI cells were treated overnight with or without IWP-2 (2.5 µM), and the WCLs were subjected to immunoblotting with the indicated antibodies. (**D, E**) IWP-2 increased the cell surface levels of endogenous FZD5/8 but not FZD4. HEK293A WT cells were treated with or without IWP-2 and analyzed by flow cytometry to determine the FZD5/8 and FZD4 levels on the cell surface with anti-FZD5/8 (2919) or anti-FZD4 (5028) monoclonal antibodies. APC, allophycocyanin. (**F, G**) *ZNRF3/RNF43* double knockout increased the cell surface levels of endogenous FZD5/8 but not FZD4. The cell surface levels of FZD5/8 and FZD4 in WT or ZRDKO cells were analyzed by flow cytometry. (**H**) Wnt-induced FZD5 degradation is dependent on endogenous ZNRF3 and RNF43. WT, FZD5KI, or ZRDKO-FZD5KI (*ZNRF3/RNF43* double knockout cells with a V5 tag knocked in at the C-terminus of endogenous FZD5) cells were treated and analyzed as described in (**B**). (**I**) IWP-2 did not increase the endogenous FZD5 protein level in ZRDKO cells. (**J**) RSPO1 treatment increased the level of the mature form of endogenous FZD5 but not FZD7. FZD5KI and FZD7KI cells were treated with control or RSPO1 CM for 4 hr, and the WCLs were analyzed by immunoblotting with the indicated antibodies. (**K**) RSPO1 failed to increase the level of the mature form of FZD5 in ZRDKO cells. FZD5KI or ZRDKO-FZD5KI cells were treated and tested as described in (**J**). (**L**) RSPO1 treatment restored Wnt3a and Wnt5a induced membrane FZD5 degradation. HEK293A FZD5KI cells were incubated with EZ-link Sulfo-NHS-SS-Biotin to label membrane protein, then treated with control, Wnt3a, Wnt5a CM, or Wnt3a/Wnt5a with RSPO1 CM. Membrane proteins were bound by NeutrAvidin beads and eluted for immunoblotting. (**M**) IWP-2 and RSPO1 treatments similarly increased the FZD5/8 levels on the cell surface. WT cells were treated with IWP-2 overnight or RSPO1 for 4 hr, followed by flow cytometry analysis. (**N, O**) RSPO1 treatment did not increase the cell surface levels of FZD5/8 in ZRDKO cells or FZD4 in WT cells. HEK293A ZRDKO or WT cells were treated with control or RSPO1 CM for 4 hr and subjected to flow cytometry analysis with anti-FZD5/8 or anti-FZD4 monoclonal antibodies.

The online version of this article includes the following source data and figure supplement(s) for figure 2:

**Source data 1.** Raw unedited blots for *Figure 2*.

**Source data 2.** Uncropped and labeled blots for *Figure 2*.

**Figure supplement 1.** IWP-2 increased the cell surface levels of endogenous FZD5/8 in several cell lines.

**Figure supplement 2.** Characterization of the antibodies used for flow cytometry in this study.

**Figure supplement 3.** Genetic lesions in ZRDKO cells.

## ZNRF3/RNF43 promotes the degradation of internalized FZD5 but is dispensable for Wnt-induced FZD5 internalization

We next investigated the role of ZNRF3/RNF43 in the process of Wnt-induced FZD5/8 endocytosis and degradation. We found that Wnt3a or Wnt5a CM treatment reduced the cell surface level of V5-FZD5 in ZRDKO cells, as determined by flow cytometry (*Figure 4A*), although they had little effect on the total protein level of FZD5 (*Figure 4B*), suggesting that Wnt induces FZD5 internalization independent of ZNRF3/RNF43. We further performed an immunostaining assay to visualize the cellular trafficking route of internalized FZD5 after Wnt3a and Wnt5a treatment for various durations. The results revealed that a significant amount of FZD5 was internalized into the intracellular vesicles in both WT and ZRDKO cells after Wnt3a and Wnt5a treatment for one hour, and the FZD5 signals gradually diminished in WT but not ZRDKO cells after Wnt3a and Wnt5a treatment for two and four hours (*Figure 4C*, *Figure 4—figure supplement 1A*), indicating that ZNRF3/RNF43 is required for the degradation of internalized FZD5 but not for Wnt-induced FZD5 internalization. Furthermore, cotreatment with RSPO1 prevented FZD5 degradation but had little effect on FZD5 internalization induced by Wnt3a and Wnt5a (*Figure 4D*, *Figure 4—figure supplement 1B*). Under Wnt5a treatment, internalized FZD5 colocalized with either EEA1, an early endosomal marker, or LAMP1, a lysosomal marker, in WT cells, whereas this colocalization was significantly reduced in ZRDKO cells (*Figure 4E–H*). These results suggest that ZNRF3/RNF43 promotes endolysosomal transport and degradation of internalized FZD5, whereas in ZRDKO cells, internalized FZD5 may be recycled back to the plasma membrane for reuse.

## Reevaluation of the function of ZNRF3/RNF43 in Wnt signaling

Previous studies have suggested that ZNRF3/RNF43 degrades FZD receptors independently of Wnt ligands (*Farnhammer et al., 2023*; *Jiang et al., 2015*; *Tsukiyama et al., 2015*). However, our finding that Wnt induces ZNRF3/RNF43-dependent FZD5/8 degradation indicates that the functions of ZNRF3/RNF43 in FZD endocytosis and degradation are modulated by Wnt ligands. These results promoted us to reevaluate the function of ZNRF3/RNF43 in Wnt signaling. Although HEK293A cells stably expressing V5-FZD5 or V5-FZD7 presented significantly higher FZD protein levels on the cell surface than ZRDKO cells did, as measured by flow cytometry with an anti-pan-FZD antibody (*Figure 5A*,

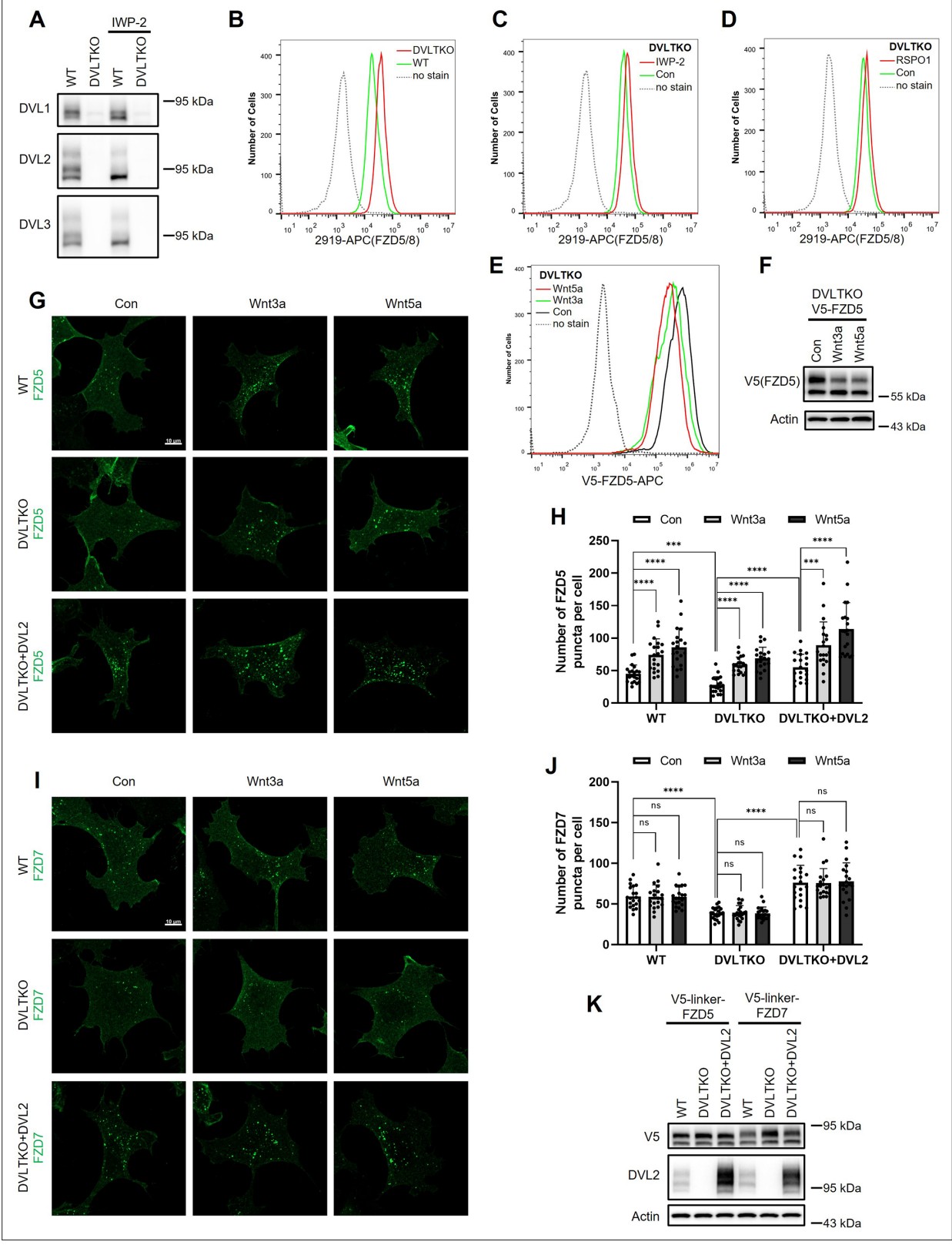

**Figure 3.** DVL proteins participate in ligand-independent FZD protein endocytosis but are not required for Wnt-induced FZD5/8 endocytosis or degradation. (**A**) Validation of HEK293A DVLTKO cells by immunoblotting analysis with the indicated anti-DVL antibodies. (**B**) Triple knockout of *DVL1/2/3* increased FZD5/8 levels on the cell surface. WT and DVLTKO cells were analyzed by flow cytometry with an anti-FZD5/8 monoclonal antibody. (**C, D**) IWP-2 (**C**) or RSPO1 (**D**) treatment increased FZD5/8 levels on the cell surface of DVLTKO cells. DVLTKO cells were treated with IWP-2 overnight

*Figure 3 continued on next page*

*Figure 3 continued*

or RSPO1 for 4 hr, followed by flow cytometry analysis. (**E**) Wnt3a and Wnt5a treatment reduced the cell surface levels of V5-FZD5 in DVLTKO cells. HEK293A DVLTKO cells stably expressing V5-FZD5 were treated with control, Wnt3a CM, or Wnt5a CM for 4 hr and analyzed via flow cytometry with an anti-V5 antibody. (**F**) Wnt3a and Wnt5a treatment decreased the mature form of V5-FZD5 in DVLTKO cells. The cells in (**E**) were treated, and the WCLs were analyzed by immunoblotting with the indicated antibodies. (**G, I**) Triple knockout of *DVL1/2/3* reduced ligand-independent FZD5 and FZD7 endocytosis, but had no effect on Wnt3a or Wnt5a induced FZD5 endocytosis. And DVLTKO cells re-expressing DVL2 rescued decreased FZD5 and FZD7 endocytosis caused by *DVL1/2/3* triple knockout. HEK293A cells stably expressing V5-linker-FZD5 or V5-linker-FZD7 were first incubated with an anti-V5 antibody, and the cells were treated with control, Wnt3a, or Wnt5a CM for 1 hr and subjected to immunofluorescence analysis. The cells were treated with IWP-2 overnight prior to treatment with conditioned medium. Scale bars, 10 μm. (**H, J**) Quantification of the number of FZD5 (**G**) or FZD7 (**I**) puncta in WT, DVLTKO, and DVLTKO + DVL2 cells (mean ± SD, n=20 cells per group). ns, no significant difference; ***p<0.001; ****p<0.0001. (**K**) WCLs from WT, DVLTKO, and DVLTKO + DVL2 cells stably expressing V5-linker-FZD5 or V5-linker-FZD7 were analyzed by immunoblotting.

The online version of this article includes the following source data and figure supplement(s) for figure 3:

**Source data 1.** Raw unedited blots for *Figure 3*.

**Source data 2.** Uncropped and labeled blots for *Figure 3*.

**Figure supplement 1.** Genetic lesions in DVLTKO cells.

*Figure 2—figure supplement 2*), these cells presented markedly lower cytosolic β-catenin levels than ZRDKO cells did (*Figure 5B*). The increase in DVL proteins phosphorylation and cytosolic β-catenin observed in *ZNRF3/RNF43* double knockout cells could be reversed by IWP-2 treatment, whereas IWP-2 had a minimal effect on DVL phosphorylation in cells overexpressing V5-FZD5 or V5-FZD7 (*Figure 5C and D*). These findings suggest that elevated Wnt signaling in ZRDKO cells cannot be solely attributed to increased FZD protein levels on the cell surface.

Furthermore, RNF130 and RNF150 are two structurally similar transmembrane E3 ubiquitin ligases compared to ZNRF3 and RNF43. These four E3 ubiquitin ligases were overexpressed into ZRDKO cells, and only ZNRF3 or RNF43 decreased the cytosolic β-catenin levels in ZRDKO cells (*Figure 5E*). However, expressing RNF43 or RNF150 in HEK293A cells stably expressing V5-FZD5 or V5-FZD7 efficiently decreased the levels of the mature forms of FZD5 and FZD7 (*Figure 5F*). Each of these E3 ligases reduced FZD protein levels on the cell surface of the ZRDKO cells to a comparable degree, as determined by flow cytometry with an anti-pan-FZD antibody (*Figure 5G*). Furthermore, each of the four E3 ligases also reduced FZD5/8 levels on the cell surface in ZRDKO cells, with IWP-2 partially restoring the effects of ZNRF3 and RNF43 but not those of RNF130 or RNF150 (*Figure 5H*). Taken together, these results suggest that overexpressing a transmembrane E3 ubiquitin ligase may nonspecifically degrade FZD receptors and that the specific ability of ZNRF3 and RNF43 to degrade FZD is dependent on Wnt ligands.

## ZNRF3/RNF43 specifically inhibits FZD5/8-mediated Wnt signaling

Given that Wnt specifically induces FZD5/8 endocytosis and degradation and that ZNRF3/RNF43-mediated FZD5/8 degradation relies on Wnt ligands, we hypothesized that ZNRF3/RNF43 specifically inhibits FZD5/8-mediated Wnt signaling. To test this hypothesis, we knocked out both *FZD5* and *FZD8* in ZRDKO cells (ZRDKO-FZD5/8DKO) via the CRISPR/Cas9 system and verified the results via genomic sequencing and flow cytometry (*Figure 6A*, *Figure 6—figure supplement 2A*). Immunoblotting revealed that the depletion of *FZD5/8* in ZRDKO cells dramatically reduced cytosolic β-catenin levels (*Figure 6B*), which were restored by re-expressing FZD5 but not FZD7 (*Figure 6C*). Furthermore, stably expressing FZD5 but not FZD7 further elevated cytosolic β-catenin levels in ZRDKO cells (*Figure 6D*). Additionally, stable expression of RNF43 in HEK293A or U2OS cells had a minimal effect on Wnt3a-induced β-catenin accumulation; interestingly, coexpressing FZD5 but not FZD7 in RNF43-expressing cells increased the inhibitory effect of RNF43 on Wnt3a-induced β-catenin accumulation (*Figure 6E and F*, *Figure 6—figure supplement 1A and B*).

We also generated *FZD5/8* double knockout HEK293A cells (FZD5/8 DKO) via the CRISPR/Cas9 system and verified them via genomic sequencing and flow cytometry analysis (*Figure 6G*, *Figure 6—figure supplement 2B*). Immunoblotting revealed that *FZD5/8* double knockout had little effect on Wnt3a-induced cytosolic β-catenin accumulation but dramatically abolished RSPO1-induced cytosolic β-catenin accumulation (*Figure 6H and I*), which was reversed by re-expressing FZD5 or FZD8 in FZD5/8 DKO cells (*Figure 6J and K*).

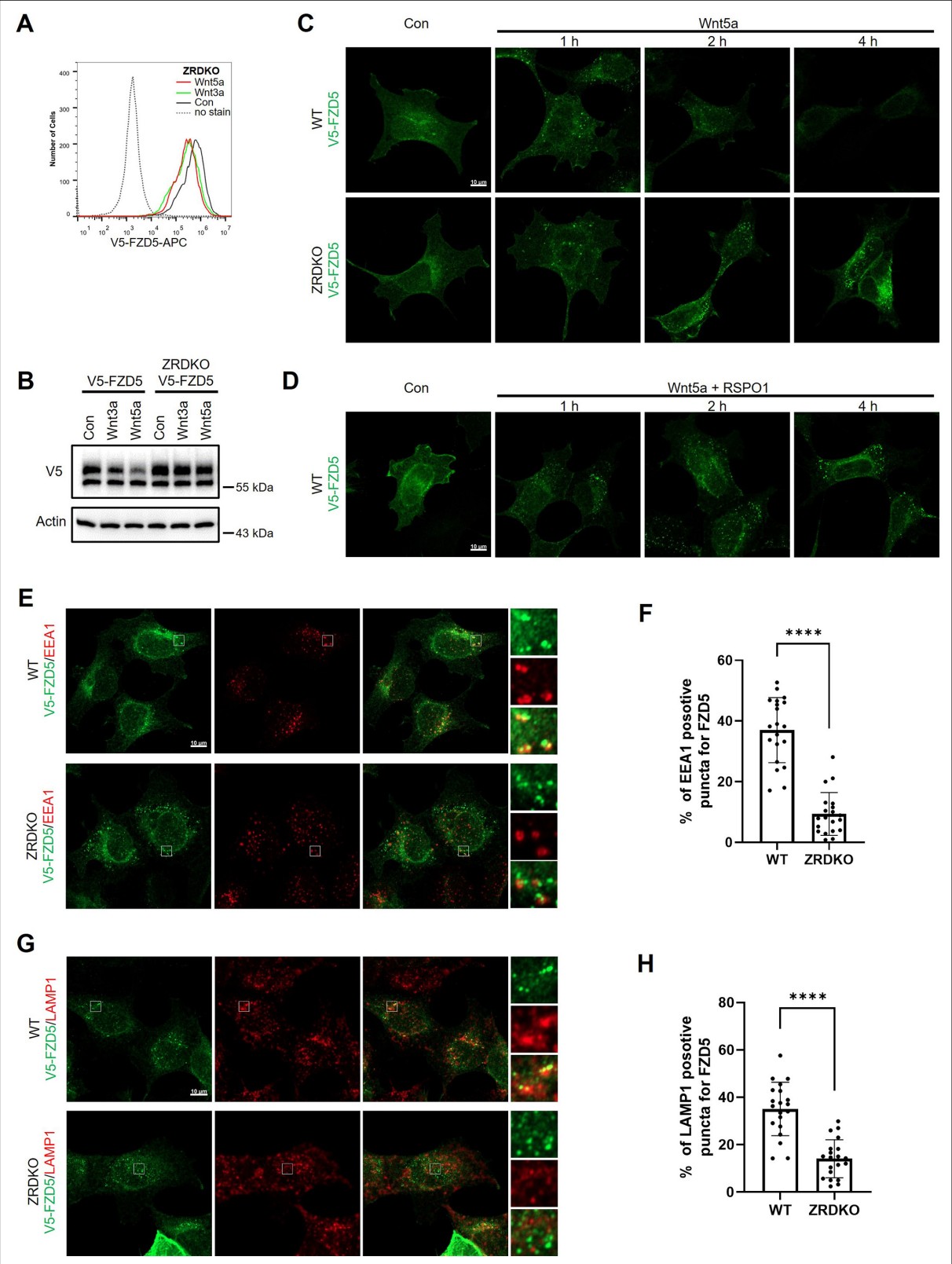

**Figure 4.** ZNRF3/RNF43 is required for the degradation of internalized FZD5 but is dispensable for Wnt-induced FZD5 internalization. (**A**) Wnt3a and Wnt5a induced V5-FZD5 endocytosis in ZRDKO cells. (**B**) Wnt3a and Wnt5a induced V5-FZD5 degradation in WT but not ZRDKO cells. (**C**) Wnt5a induced FZD5 internalization in both WT and ZRDKO cells, and internalized FZD5 gradually diminished in WT but not ZRDKO cells. WT or ZRDKO cells stably expressing V5-FZD5 were treated with control or Wnt5a CM for the indicated times and analyzed by immunostaining. The cells were treated with

*Figure 4 continued on next page*

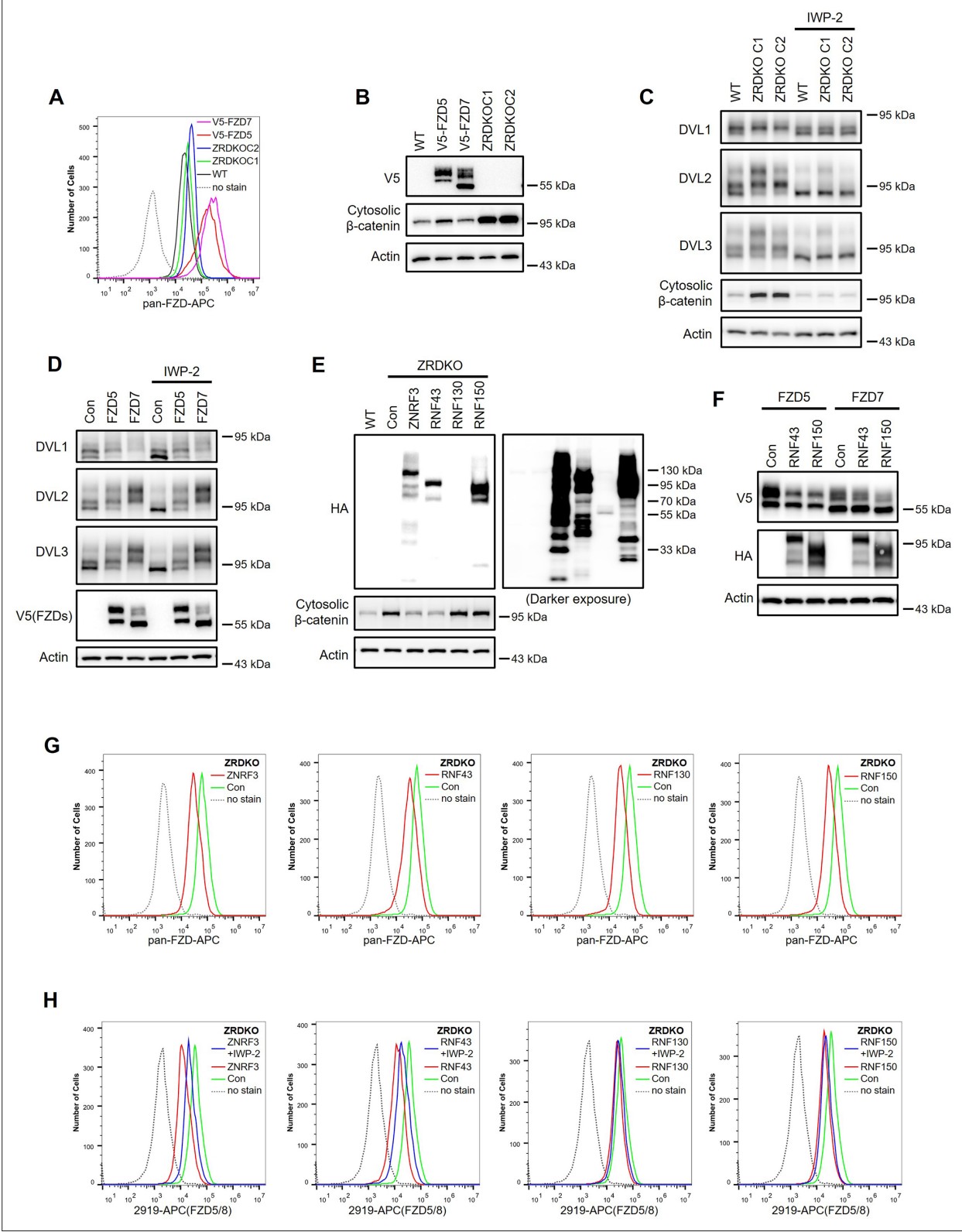

**Figure 5.** ZNRF3/RNF43-mediated inhibition of Wnt signaling cannot be explained by simply regulating FZD levels on the cell surface. (**A**) Flow cytometry analysis of FZD levels on the cell surface of WT HEK293A, ZRDKO (ZRDKO C1/2), and HEK293A cells stably expressing V5-FZD5/7 with an anti-pan-FZD monoclonal antibody. FZD levels on the cell surface were greater in ZRDKO cells than in WT cells but much lower than those in V5-FZD5/7-expressing cells. (**B**) Cytosolic β-catenin levels were significantly lower in V5-FZD5/7-expressing cells than in ZRDKO cells. (**C**) *ZNRF3/RNF43* double

*Figure 5 continued on next page*

*Figure 4 continued*

IWP-2 overnight prior to treatment with CM. Scale bars, 10 μm. (**D**) Cotreatment with RSPO1 prevented FZD5 degradation but had little effect on FZD5 internalization induced by Wnt5a. The cells were treated with IWP-2 overnight prior to treatment with CM. Scale bars, 10 μm. (**E**) Compared with those in WT cells, fewer V5-FZD5 puncta colocalized with the early endosomal marker EEA1 in ZRDKO cells. WT or ZRDKO cells stably expressing V5-FZD5 were treated with control or Wnt5a CM for 2 hr and then analyzed by immunostaining. The cells were treated with IWP-2 overnight prior to treatment with CM. Scale bars, 10 μm. (**F**) Quantification of the percentage of V5-FZD5 puncta colocalized with EEA1 in WT and ZRDKO cells (mean ± SD, n=20 cells per group). ****p<0.0001. (**G**) Compared with those in WT cells, fewer V5-FZD5 puncta colocalized with the lysosomal marker LAMP1 in ZRDKO cells. WT or ZRDKO cells stably expressing V5-FZD5 were treated with control or Wnt5a CM for 2 hr and then analyzed by immunostaining. The cells were treated with IWP-2 overnight prior to treatment with CM. Scale bars, 10 μm. (**H**) Quantification of the percentage of V5-FZD5 puncta colocalized with LAMP1 in WT and ZRDKO cells (mean ± SD, n=20 cells per group). ****p<0.0001.

The online version of this article includes the following source data and figure supplement(s) for figure 4:

**Source data 1.** Raw unedited blots for *Figure 4*.

**Source data 2.** Uncropped and labeled blots for *Figure 4*.

**Figure supplement 1.** ZNRF3/RNF43 is required for the degradation of Wnt3a induced internalized FZD5.

Collectively, these findings indicate that RSPO1 enhances Wnt signaling primarily through FZD5/8, whereas ZNRF3/RNF43 specifically inhibits Wnt signaling mediated by FZD5/8.

## Wnt induces FZD5 and RNF43 interaction

As Wnt ligands induce ZNRF3/RNF43-dependent FZD5/8 degradation, we hypothesized that these ligands may also promote interactions between FZD5/8 and ZNRF3/RNF43 at the plasma membrane. To test this hypothesis, we utilized a proximity biotin labeling technique, which is well suited for detecting inducible and transient interactions (*Branon et al., 2018*; *Li et al., 2023*). To minimize interference from endogenous ZNRF3/RNF43, we employed HEK293A ZRDKO cells stably expressing V5-FZD5-mTB or V5-FZD7-mTB (with miniTurbo biotin ligase fused to the carboxyl terminus of FZD5 or FZD7) alongside RNF43ΔRING-HA (lacking the enzymatic RING domain of RNF43). We confirmed the expression of FZD5/7-mTB and RNF43ΔRING, as well as the biotinylation efficiency, through immunoblotting (*Figure 7A*). We subsequently conducted proximity biotin labeling assays with or without Wnt3a, Wnt5a, or IWP-2 treatment. Both Wnt3a and Wnt5a significantly increased the biotinylation of RNF43ΔRING in FZD5-mTB-expressing cells but not in FZD7-mTB-expressing cells (*Figure 7B*). Additionally, IWP-2 treatment reduced the basal biotinylation of RNF43ΔRING in FZD5-mTB-expressing cells (*Figure 7B*). We further replicated the biotin-based proximity labeling assay in DVLTKO cells. The results showed that DVL proteins did not participate in Wnt3a or Wnt5a-induced FZD5 and RNF43 interaction (*Figure 7—figure supplement 1A and B*). These results suggest that Wnt specifically induces DVL proteins independent of interaction between FZD5 and RNF43 at the plasma membrane.

## Discussion

Our results demonstrate that Wnt signaling employs a strategy analogous to that of receptor–tyrosine kinase pathways, where ligands bind to and activate receptors while simultaneously inducing receptor endocytosis and degradation to modulate signaling strength and duration (*Alexander, 1998*; *Sorkin and Waters, 1993*). Our results also illuminate the mechanism by which RSPO-ZNRF3/RNF43 regulates Wnt-induced FZD5/8 endocytosis and degradation and its influence on Wnt signaling. In the presence of ZNRF3/RNF43, Wnt binds to FZD5/8, forming a complex with ZNRF3/RNF43 and facilitating FZD5/8 endocytosis and subsequent lysosomal degradation, and this process occurs independently of DVL proteins. In the absence of ZNRF3/RNF43, Wnt still induces FZD5/8 endocytosis; however, endosomes harboring FZD5/8 fail to merge with lysosomes, leading to the accumulation of mature FZD5/8 within the cell, which may be recycled back to the plasma membrane for reuse to sustain signaling activity (*Figure 7C*).

There are 10 FZD proteins in mammals that are classified into five subfamilies on the basis of their sequence similarity. Among them, FZD5/8 was reported to have higher signaling activity and broader Wnt binding and response spectrum (*Martin et al., 2022*; *Wang et al., 2016*). A recent study suggested that FZD5 undergoes maturation through specific binding to cholesterol (*Zheng et al., 2022*). Our current study suggests that Wnt specifically induces FZD5/8 endocytosis and degradation, adding a new feature to the FZD5/8 subfamily. Wnt-induced FZD5/8 endocytosis relies on the CRD

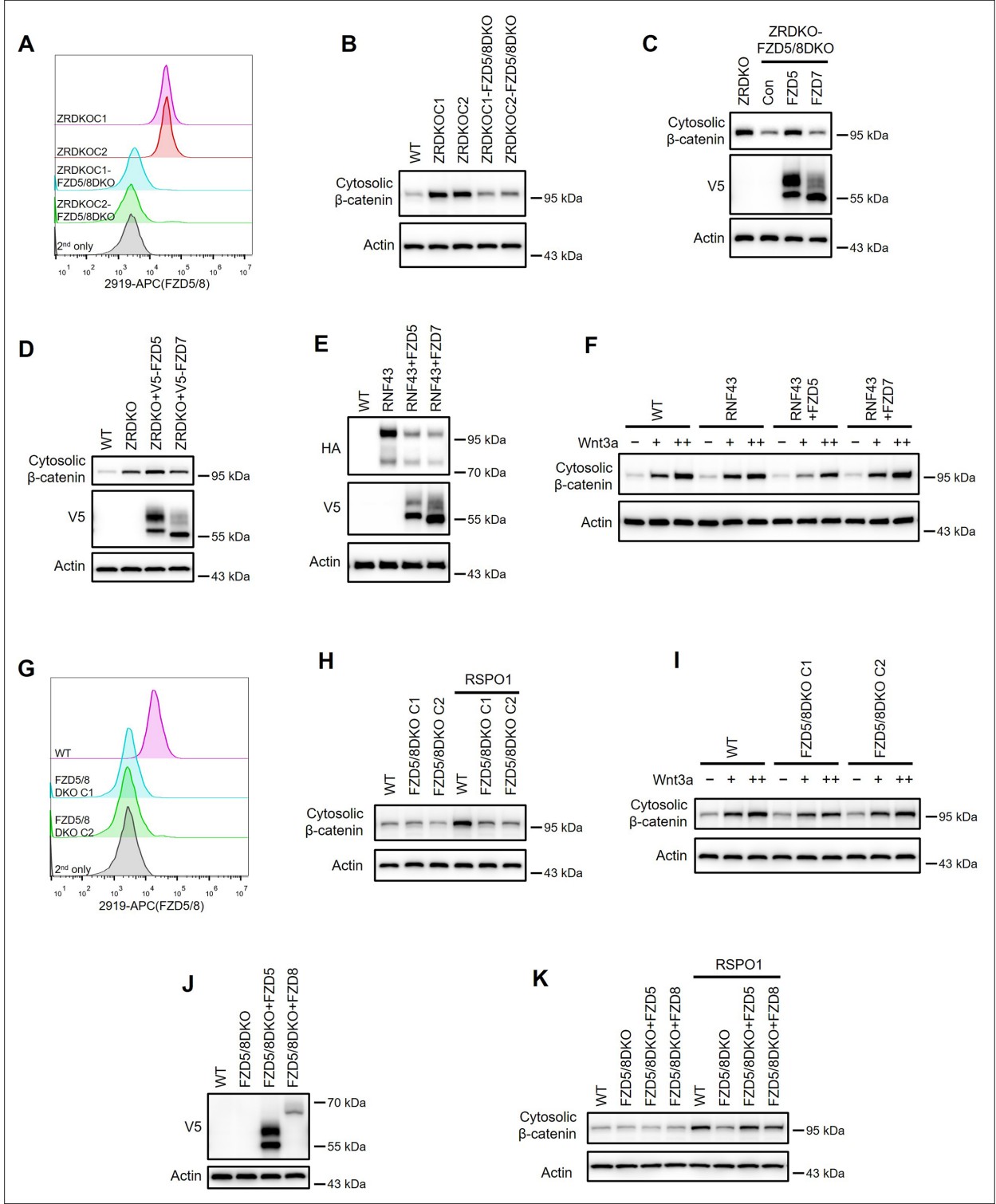

**Figure 6.** ZNRF3/RNF43 specifically inhibits FZD5/8-mediated Wnt signaling, whereas RSPO1 specifically potentiates FZD5/8-mediated Wnt signaling. (**A**) Flow cytometry analysis of FZD5/8 levels on the cell surface in ZRDKO and ZRDKO-FZD5/8DKO cells with an anti-FZD5/8 antibody. (**B**) Depletion of FZD5/8 diminished the increase in cytosolic β-catenin levels in ZRDKO cells. Cytosolic fractions from the indicated cells were analyzed by immunoblotting. (**C**) Re-expression of V5-FZD5, but not V5-FZD7, restored cytosolic β-catenin levels in ZRDKO-FZD5/8DKO cells. (**D**) Overexpression of V5-FZD, but not V5-FZD7, further elevated cytosolic β-catenin levels in ZRDKO cells. (**E, F**) FZD5, but not FZD7, enhanced the inhibitory effect of RNF43 on Wnt3a signaling. HEK293A cells stably expressing RNF43-HA alone or with V5-FZD5 or V5-FZD7 were treated with increasing doses of Wnt3a for 2 hr, and the cytosolic fractions were analyzed by immunoblotting. (**G**) Flow cytometry analysis of cell surface FZD5/8 levels in WT or FZD5/8 DKO cells.

*Figure 6 continued on next page*

*Figure 6 continued*

(**H, I**) *FZD5/8* double knockout abolished RSPO1-induced cytosolic β-catenin accumulation but had little effect on Wnt3a-induced increases in cytosolic β-catenin levels. The cells were treated with RSPO1 CM for 4 hr (**H**) or increasing doses of Wnt3a for 2 hr (**I**), and the cytosolic fractions were analyzed by immunoblotting. (**J**) WCLs from WT, FZD5/8 DKO and FZD5/8 DKO cells stably expressing V5-FZD5 or V5-FZD8 were analyzed by immunoblotting. (**K**) *FZD5/8* double knockout abolished RSPO1-induced cytosolic β-catenin accumulation, which was restored by re-expressing V5-FZD5 or V5-FZD8.

The online version of this article includes the following source data and figure supplement(s) for figure 6:

**Source data 1.** Raw unedited blots for *Figure 6*.

**Source data 2.** Uncropped and labeled blots for *Figure 6*.

**Figure supplement 1—source data 1.** Raw unedited blots for *Figure 6—figure supplement 1*.

**Figure supplement 1—source data 2.** Uncropped and labeled blots for *Figure 6—figure supplement 1*.

**Figure supplement 1.** ZNRF3/RNF43 specifically regulates FZD5/8-mediated Wnt signaling.

**Figure supplement 2.** Genetic lesions in ZRDKO-FZD5/8DKO and FZD5/8DKO cells.

domain and is dispensable for the FZD5/8 intracellular carboxyl terminus. It is possible that the Wnt-FZD5/8 complex recruits another unidentified membrane protein that promotes the internalization of the whole complex. Wnt3a and Wnt5a bind to FZD5/8, FZD4, and FZD7 with similar affinities (*Li et al., 2023*) and specifically induce FZD5/8 but not other FZDs endocytosis. Therefore, further characterization and comparison of the structural differences between Wnt-bound CRDs from different FZDs and identification of specific binding proteins of the Wnt-FZD5/8 complex will shed light on the underlying mechanism. Although we showed that Wnt3a, Wnt5a, and endogenously expressed Wnts in 293 A cells specifically induce FZD5/8 endocytosis, we cannot exclude the possibility that other Wnts or ligands may induce the endocytosis and degradation of other FZDs in certain biological contexts.

Our results also shed light on the function of DVL proteins in FZD endocytosis. We showed that FZD receptors undergo ligand-independent constant endocytosis that relies on DVL. However, Wnt-induced FZD5/8 endocytosis and degradation occur in the absence of DVL proteins. A previous study reported that DVL proteins interact with ZNRF3/RNF43 and may transport ZNRF3/RNF43 to FZD, thus facilitating FZD endocytosis and degradation (*Jiang et al., 2015*). Notably, these results are largely based on flow cytometry assays, which can detect only the internalization but not the degradation of FZDs. Furthermore, they used a ZNRF3/RNF43ΔDIR-DEP fusion protein and showed that expressing this fusion protein efficiently removed FZD proteins from the cell surface. Notably, the DEP domain of DVL also interacts with FZDs, and the ZNRF3/RNF43ΔDIR-DEP fusion protein may function as an anti-FZD 'PROTAC' to degrade FZDs. Our results indicated that Wnt-induced FZD5/8 degradation requires ZNRF3/RNF43 but is dispensable for DVL proteins, suggesting that the interaction between DVL and ZNRF3/RNF43 is not required for this process. We cannot exclude the possibility that ZNRF3/RNF43 may regulate ligand-independent FZD endocytosis and degradation under certain circumstances that may rely on the DVL-ZNRF/RNF43 interaction. It is also possible that DVL interacts with ZNRF3/RNF43 to regulate ZNRF3/RNF43 endocytosis. Our results also revealed that overexpressing the membrane RING-type E3 ligases RNF130 and RNF150 reduced membrane FZD levels without affecting cytosolic β-catenin levels, suggesting potential nonspecific effects of ZNRF3/RNF43 in overexpression assays.

Our results suggest that ZNRF3/RNF43 regulates the degradation of internalized FZD5/8 but is not required for Wnt-induced FZD5/8 internalization. Wnt can still induce FZD5 internalization in the absence of ZNRF3/RNF43, and internalized FZD5 accumulates on intracellular vesicles in both ZRDKO cells and WT cells treated with RSPO1. It is well established that internalized membrane proteins are sorted in endosomes either for recycling or for lysosomal degradation (*Cullen and Steinberg, 2018*). The ubiquitination of intracellular lysine residues serves as a typical sorting signal that can be recognized by the endosomal sorting complexes required for transport (ESCRT) and sorted for lysosomal degradation (*Cullen and Steinberg, 2018*). It is conceivable that ZNRF3/RNF43 ubiquitinates FZD5/8 during Wnt-induced FZD5/8 endocytosis and that the internalized ubiquitinated FZD5/8 is then sorted for lysosomal degradation. In the absence of ZNRF3/RNF43 or with RSPO treatment, internalized FZD5/8 is not ubiquitinated and thus is sorted for recycling. However, further investigations are needed to elucidate the detailed mechanism of the intracellular vesicular trafficking of internalized FZDs.

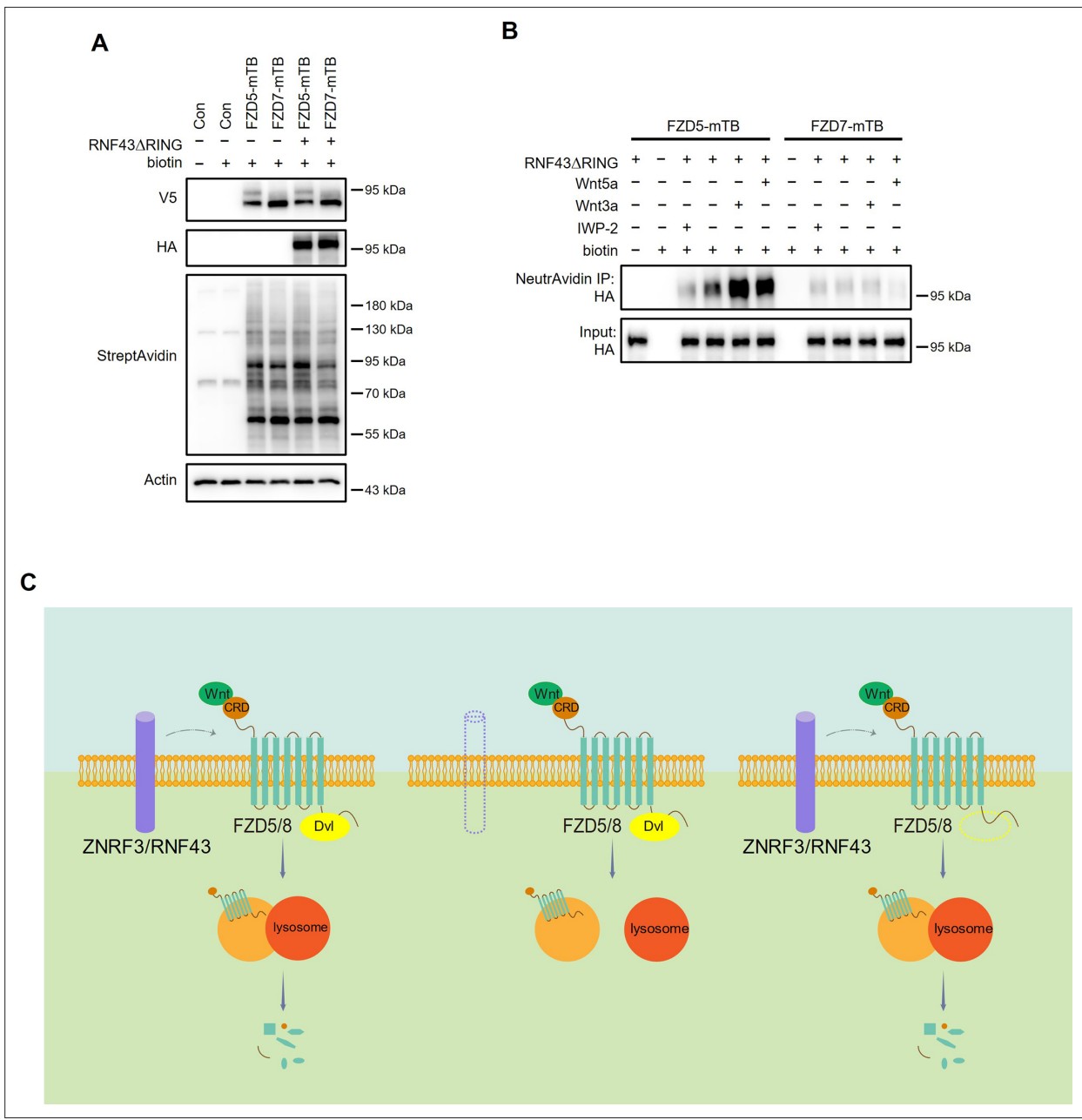

**Figure 7.** Wnt3a and Wnt5a induce the FZD5–RNF43 interaction. (**A**) Verification of protein expression and biotinylation efficiency in HEK293A ZRDKO cells stably expressing V5-FZD5-mTB or V5-FZD7-mTB alone or together with RNF43ΔRING-HA. The cells were treated with or without biotin (200 μM) and analyzed by immunoblotting with the indicated antibodies. (**B**) Wnt3a or Wnt5a induces interaction between FZD5 and RNF43ΔRING but not FZD7. The cells in (**A**) were treated with IWP-2, control, Wnt3a or Wnt5a conditional medium with or without biotin as indicated, and the WCLs were precipitated with NeutrAvidin agarose beads, followed by immunoblotting analysis. (**C**) Schematic model of Wnt-induced FZD5/8 endocytosis and degradation. In the presence of ZNRF3/RNF43, Wnt binds to FZD5/8 and forms a complex with ZNRF3/RNF43, leading to FZD5/8 endocytosis and subsequent lysosomal degradation. This process is not affected by the absence of DVL proteins. In the absence of ZNRF3/RNF43, Wnt still induces FZD5/8 endocytosis; however, endosomes containing FZD5/8 do not merge with lysosomes, resulting in the accumulation of mature FZD5/8 intracellularly.

The online version of this article includes the following source data and figure supplement(s) for figure 7:

**Source data 1.** Raw unedited blots for *Figure 7*.

**Source data 2.** Uncropped and labeled blots for *Figure 7*.

**Figure supplement 1.** Wnt3a and Wnt5a induce the FZD5–RNF43 interaction in DVLTKO cells.

**Figure supplement 1—source data 1.** Raw unedited blots for *Figure 7—figure supplement 1*.

**Figure supplement 1—source data 2.** Uncropped and labeled blots for *Figure 7—figure supplement 1*.

*Figure 5 continued*

knockout-induced DVL phosphorylation and cytosolic β-catenin accumulation depend on endogenous Wnt proteins. WT or ZRDKO cells were treated with or without IWP-2 and analyzed by immunoblotting. (**D**) Overexpressing V5-FZD5 or V5-FZD7 resulted in Wnt-independent DVL phosphorylation. V5-FZD5/7-expressing cells were treated with or without IWP-2, and the WCLs were analyzed by immunoblotting. (**E**) Expression of ZNRF3 or RNF43, but not RNF130 or RNF150, reversed the increase in cytosolic β-catenin levels in ZRDKO cells. (**F**) Overexpression of RNF43 or RNF150 reduced the levels of mature forms of V5-FZD5 or V5-FZD7. WCLs from HEK293A cells expressing V5-FZD5 or V5-FZD7 alone or together with RNF43-HA or RNF150-HA were analyzed by immunoblotting. (**G**) Overexpression of ZNRF3, RNF43, RNF130, or RNF150 in ZRDKO cells reduced FZD levels on the cell surface, as measured by flow cytometry with an anti-pan-FZD antibody. (**H**) Overexpression of ZNRF3, RNF43, RNF130, or RNF150 in ZRDKO cells reduced FZD5/8 levels on the cell surface. IWP-2 partially reversed the reduction in FZD5/8 levels induced by ZNRF3 or RNF43 but not by RNF130 or RNF150. The cells in (**E**) were treated with or without IWP-2 overnight and analyzed by flow cytometry with an anti-FZD5/8 monoclonal antibody.

The online version of this article includes the following source data for figure 5:

**Source data 1.** Raw unedited blots for *Figure 5*.

**Source data 2.** Uncropped and labeled blots for *Figure 5*.

Our results suggest that RSPO-ZNRF3/RNF43 specifically targets Wnt-bound FZD5/8, as *FZD5/8* double knockout largely abolished Wnt/β-catenin signaling activity in ZRDKO cells or WT cells treated with RSPO1. Our results provide an underlying mechanism for a recent study in which the authors reported that FZD5, but not the other FZDs, is crucial for the growth of RNF43-mutated pancreatic tumors and colorectal tumor organoids (*Steinhart et al., 2017*). This mechanism may be crucial for stem cell self-renewal and tissue homeostasis. RSPO proteins, known as stem cell factors, are present in stem cell niches (*de Lau et al., 2011*; *Yan et al., 2017*), while ZNRF3/RNF43 are also expressed in stem cells (*Koo et al., 2012*; *Sato et al., 2011*; *Yan et al., 2017*). FZD5 is expressed in adult stem cells and is essential for their function (*Alsaadi et al., 2023*; *Harada et al., 2021*; *Nabhan et al., 2023*; *Yan et al., 2017*; *Zheng et al., 2022*).

In summary, our study introduces a novel concept in Wnt signaling regulation at the FZD receptor level, clarifies the mechanisms by which RSPO-ZNRF3/RNF43 regulates Wnt signaling, and offers new insights into Wnt signaling-related regenerative medicine and cancer therapy.

## Materials and methods

**Key resources table**

| Reagent type (species) or resource | Designation | Source or reference | Identifiers | Additional information |
|---|---|---|---|---|
| Antibody | Rabbit monoclonal anti-HA | Cell Signaling Technology | Cat# 3724; RRID:AB_1549585 | WB(1:1000) |
| Antibody | Rabbit monoclonal anti-V5 | Cell Signaling Technology | Cat# 13202; RRID:AB_2687461 | WB(1:1000) IF(1:1000) FACS(1:500) |
| Antibody | Mouse monoclonal anti-V5 | Cell Signaling Technology | Cat# 80076; RRID:AB_2920661 | IF(1:1000) |
| Antibody | Rabbit polyclonal anti-Beta Catenin | Proteintech | Cat# 51067–2-AP; RRID:AB_2086128 | WB(1:10000) |
| Antibody | Rabbit polyclonal anti-DVL1 | Proteintech | Cat# 27384–1-AP; RRID:AB_2880859 | WB(1:1000) |
| Antibody | Mouse monoclonal anti-DVL2 | Proteintech | Cat# 67105–1-Ig; RRID:AB_2882409 | WB(1:2000) |
| Antibody | Rabbit polyclonal anti-DVL3 | Proteintech | Cat# 13444–1-AP; RRID:AB_2093451 | WB(1:2000) |
| Antibody | Rabbit monoclonal anti-ACTB | ABclonal | Cat# AC026; RRID:AB_2768234 | WB(1:10,000) |
| Antibody | Rabbit monoclonal anti-LAMP1 | Cell Signaling Technology | Cat# 9091; RRID:AB_2687579 | IF(1:500) |

*Continued on next page*

*Continued*

| Reagent type (species) or resource | Designation | Source or reference | Identifiers | Additional information |
|---|---|---|---|---|
| Antibody | Mouse monoclonal anti-EEA1 | Cell Signaling Technology | Cat# 48453, RRID:AB_2920538 | IF(1:500) |
| Antibody | Alexa Fluor 488 AffiniPure Donkey anti-Rabbit IgG (H+L) | Yeasen Biotech | Cat# 34206ES60; RRID:AB_2909605 | IF(1:400) |
| Antibody | Alexa Fluor 488 AffiniPure Donkey Anti-Mouse IgG(H+L) | Yeasen Biotech | Cat# 34106ES60; RRID:AB_2920874 | IF(1:400) |
| Antibody | Alexa Fluor 647 AffiniPure Donkey anti-Rabbit IgG (H+L) | Yeasen Biotech | Cat# 34213ES60 | IF(1:400) |
| Antibody | Alexa Fluor 647 AffiniPure Donkey Anti-Mouse IgG(H+L) | Yeasen Biotech | Cat# 34113ES60 | IF(1:400) |
| Antibody | Allophycocyanin-AffiniPure F(ab')2 Fragment Goat Anti-Rabbit IgG (H+L) | Jackson ImmunoResearch Labs | Cat# 111-136-144; RRID:AB_2337987 | FACS(1:200) |
| Antibody | Allophycocyanin-AffiniPure F(ab')2 Fragment Goat Anti-Mouse IgG (H+L) | Jackson ImmunoResearch Labs | Cat# 115-136-146; RRID:AB_2338651 | FACS(1:200) |
| Antibody | Peroxidase-AffiniPure Goat Anti-Rabbit IgG (H+L) | Jackson ImmunoResearch Labs | Cat# 111-035-003; RRID:AB_2313567 | WB(1:10,000) |
| Antibody | HRP-conjugated goat anti-mouse IgG (H+L) | Mabnus | Cat# GS80002 | WB(1:10,000) |
| Antibody | HRP-labeled Streptavidin | Beyotime | Cat# A0303 | WB(1:1000) |
| Antibody | Mouse monoclonal anti-pan-FZD (OMP-18R5) | *Stagg and Dupont, 2014* | N/A | FACS(1:500) |
| Antibody | Mouse monoclonal anti- FZD5/8 (2919) | *Pan et al., 2021* | N/A | FACS(1:500) |
| Antibody | Mouse monoclonal anti-FZD4 (5028) | *Sachdev Sidhu et al., 2021* | N/A | FACS(1:500) |
| Chemical compound, drug | IWP-2 | Sigma–Aldrich | Cat# I0536 | N/A |
| Chemical compound, drug | D-Biotin | Sigma–Aldrich | Cat# B4639 | N/A |
| Chemical compound, drug | Bafilomycin A1 (BA1) | Selleck | Cat# S1413 | N/A |
| Cell line (*Homo sapiens*) | HEK293T | ATCC | Cat# CRL-11268 | N/A |
| Cell line (*Homo sapiens*) | HEK293A | Procell | Cat# CL-0003 | N/A |
| Cell line (*Homo sapiens*) | Huh7 | Procell | Cat# CL-0120 | N/A |
| Cell line (*Homo sapiens*) | MCF7 | Procell | Cat# CL-0149 | N/A |
| Cell line (*Homo sapiens*) | 769 P | Procell | Cat# CL-0009 | N/A |
| Cell line (*Homo sapiens*) | U2OS | ATCC | Cat# HTB-96 | N/A |
| Cell line (*M. musculus*) | L cell | ATCC | Cat# CRL-2648 | N/A |
| Cell line (*M. musculus*) | L-Wnt3a cell | ATCC | Cat# CRL-2647 | N/A |
| Cell line (*M. musculus*) | L-Wnt5a cell | ATCC | Cat# CRL-2814 | N/A |

## Cell culture

HEK293T, HEK293A, Huh7, MCF7, 769P, U2OS, and mouse L cells stably expressing Wnt3a or Wnt5a were cultured in high-glucose Dulbecco's modified Eagle's medium (boster) supplemented with 10% fetal bovine serum (Genial) and 1% penicillin/streptomycin at 37°C in a humidified incubator (Thermo Fisher Scientific) with 5% $CO_2$. All cell lines used in this study were authenticated by STR profiling and confirmed to be free of mycoplasma contamination.

## Generation of CRISPR/Cas9 knockout (KO) cell lines

HEK293A cells were transfected with LentiCRSIPRv2 plasmids encoding Cas9 and guide RNAs via PEI transfection reagents. Forty-eight hours posttransfection, the cells were selected with puromycin (1 µg/mL). Following selection, the cells were plated at a density of 500 cells per 10 cm dish to facilitate the growth of monoclonal cell lines. Once colonies had formed, they were gently harvested by pipetting and subsequently verified via genomic DNA sequencing. Additionally, *FZD5/8* double knockout (FZD5/8 DKO) clones were confirmed via flow cytometry with an anti-FZD5/8 (2919) monoclonal antibody.

## Generation of KI cell lines

HEK293A cells were transfected with donor plasmids along with LentiCRSIPRv2 knock-in plasmids via PEI transfection reagents. Selection was performed with puromycin (1 µg/mL) 48 hr post-transfection. The cells were then further selected with G418 sulfate (1 mg/mL) for 7 days. Monoclonal cell lines were established by seeding the selected cells in 10 cm dishes. After colony formation, the cells were harvested by pipetting and verified by immunoblotting. The sequences for the knock-in sgRNAs were as follows: FZD5KI (GGTGTCCCTGTCGCACGTGT) and FZD7KI (CAGCAGCAAGGGGGGGAGAC TG).

## Plasmids

V5-FZD1-10 (in which a V5 tag was inserted after the signal peptide) was cloned and inserted into the pCDH lentiviral vector, which included a hygromycin resistance gene, and ZNRF3-HA, RNF43-HA, RNF130-HA, and RNF150-HA (in which an HA tag was inserted at the C-terminus) were cloned and inserted into the pCDH lentiviral vector, which included a puromycin resistance gene. V5- FZD5△CRD (missing amino acids 31—172) and FZD5△C (missing amino acids 524—585) were generated using full-length FZD5 as the template. FZD5CRD-FZD4, FZD5CRD-FZD7 were made by combining FZD5CRD with FZD4△CRD (missing amino acids 42—167) or FZD7△CRD (missing amino acids 45 to –169), FZD4CRD-FZD5, and FZD7CRD-FZD5 were made by combining FZD4CRD or FZD7CRD with FZD5△CRD. V5-FZD5/FZD7-mTB plasmids were generated by inserting miniTurbo biotin ligase at the C-terminus of V5-FZD5/FZD7. V5- linker-FZD5/7 was generated by inserting a linker sequence between V5 and FZD5/7 to extend the distance between the V5 tag and the FZD protein. RSPO1 was tagged with a Flag epitope at its C-terminus. RNF43ΔRING (missing amino acids 270–316) was generated using full-length RNF43 as the template. FZD5/7KI donors were constructed with the following elements sequentially connected: the left homologous arm with the stop codon replaced by a V5 tag, a P2A peptide, a G418 resistance gene, and the right homologous arm. The sequences targeted by the LentiCRISPRv2 sgRNA are detailed in the expanded view figures. All plasmids were confirmed by Sanger sequencing.

## Conditioned medium

Wnt3a- and Wnt5a-conditioned media were generated according to the ATCC's product sheet. L-Wnt3a or L-Wnt5a cells were cultured at an appropriate density in 10 cm dishes. After 4 days, the medium was collected, and the cells were replenished with fresh medium for an additional 3 days. The medium was then collected again. Medium harvested from L wild-type cells was used as a control. For RSPO1 production, HEK293T cells were transfected with an RSPO1 expression plasmid, and the culture medium was changed 24 hr posttransfection. The supernatant containing the secreted proteins was collected after an additional 24 hr. Media harvested from HEK293T cells transfected with empty pCS2+plasmids served as controls. All the collected media were subsequently centrifuged at 12000 × *g* for 20 min at 4 °C to remove cellular debris, after which their activity was tested by immunoblotting before use.

## Lentivirus packaging and stable cell line generation

HEK293T cells were seeded in 6 cm dishes to achieve approximately 50% confluency and were transfected with a total of 3 µg of plasmid, consisting of 1.5 µg of the overexpression vector, 1 µg of psPAX2, and 0.5 µg of pMD2.G. Sixteen hours posttransfection, the medium of the cells was replaced with supplemented DMEM. The virus-containing medium was harvested 48 hr and 72 hr after transfection and then filtered through 0.22 µm filters.

For the generation of stable cell lines, HEK293A or U2OS cells were seeded in 6 cm dishes. A mixture of lentivirus-containing medium with fresh medium was added to the cells. Twenty-four hours post-infection, the medium was replaced with DMEM containing 10% FBS. After an additional 24 hr, the cells were transferred to 10 cm dishes and selected with puromycin (1 µg/mL) or hygromycin (250 µg/mL) to isolate resistant cells.

## Immunoblotting

The cells were seeded in 12-well plates and treated under the indicated conditions. Following treatment, the cells were washed with PBS, lysed with 2 × protein loading buffer (125 mM Tris–HCl pH 6.8, 2% SDS, 10% glycerol, 0.004% bromophenol blue, and 8% β-mercaptoethanol) and boiled for 10 min at 95°C. For detection of FZD proteins, the cells were lysed with 0.5% NP-40 lysis buffer containing protease inhibitor cocktail for 30 minutes with gentle rotation at 4°C. The lysates were collected in 1.5 mL tubes and centrifuged at 12,000 × $g$ for 15 min at 4°C. The supernatants were then harvested and mixed with 5 × protein loading buffer. Note that the lysates intended for detecting FZD proteins should not be boiled. Cytosolic fractions were extracted using 0.015% digitonin in PBS containing a protease inhibitor cocktail for 10 minutes with gentle rotation at 4°C. The fractions were centrifuged, and the supernatants were mixed with 5 × protein loading buffer and then boiled for 10 minutes at 95°C.

Protein samples were separated by denaturing electrophoresis on 6–10% SDS–PAGE gels (Epizyme) and transferred to 0.45 µm PVDF membranes (Millipore) via a wet transfer system (Tanon). The membranes were blocked with TBST containing 5% nonfat dry milk at room temperature for 1 hr and then incubated with primary antibodies overnight at 4°C. After being washed with TBST, the membranes were incubated with secondary antibodies for 1 hr at room temperature. Following additional washes with TBST to remove unbound antibodies, specific protein bands were detected via an enhanced chemiluminescence (ECL)-immunoblotting system (GelView 6000Pro II, BLT Photon Technology).

## Flow cytometry

HEK293A cells were incubated with ice-cold PBS for 10 min with gentle rotation at 4°C, Huh7, MCF7, and 769P cells were washed with PBS once and incubated with PBS including 2 mM EDTA for 10 min at 4°C. Then cells were collected into 1.5 mL tubes via gentle pipetting. The harvested cells were subsequently centrifuged at 400 × $g$ for 2 min at 4°C. After the supernatants were removed, the pellets were resuspended and incubated with the indicated primary antibodies diluted in FACS buffer (5% goat serum in PBS) for 1 hr on ice. The cells were then washed with ice-cold PBS, centrifuged, and incubated with secondary antibodies for an additional hour on ice. Following another wash with PBS, the cells were stained with propidium iodide (PI) and analyzed via a NovoCyte Quanteon flow cytometer (Agilent). PI-negative cells are displayed in histogram plots.

## Cell surface biotinylation assay

For detecting membrane levels of endogenous FZD5, HEK293A FZD5KI cells were washed with PBS and incubated with EZ-linker sulfo-NHS-SS-biotin in PBS for 10 min on ice. Unbound reagent was removed by washing with PBS, then cells were treated with indicated medium for 2 hours at 37°C in a humidified incubator. Thereafter, the cells were washed with PBS and lysed with 0.5% NP-40 lysis buffer containing protease inhibitor cocktail for 30 min with gentle rotation at 4°C. The lysates were collected in 1.5 mL tubes and centrifuged at 12000 × $g$ for 15 min at 4°C. The supernatants were then harvested and incubated with NeutrAvidin agarose beads overnight. The beads were centrifuged at 1500 × $g$ for 2 min, washed six times with lysis buffer, and eluted with 2 × protein loading buffer containing 8% β-mercaptoethanol for 30 min at room temperature. Cell surface proteins were detected by immunoblotting.

## Immunofluorescence

The cells were seeded on circular microscope cover glasses and treated under the indicated conditions. Following treatment, the cells were fixed with 4% paraformaldehyde (PFA) for 30 min at room temperature. After fixation, the cells were permeabilized with 0.1% TX-100 in PBS for 20 min at room temperature with gentle rotation. The cells were then blocked with 5% donkey serum in PBS for 1 hr

at room temperature. Primary antibodies (diluted in 5% donkey serum) were applied overnight at 4°C. The cells were washed with PBS and incubated with fluorescently labeled secondary antibodies for 1 hr at room temperature with gentle rotation. After being washed, the cells were stained with DAPI for 5 min at room temperature and analyzed via a confocal microscope (LSM 980 with Airyscan 2, ZEISS).

To detect FZD5/7 endocytosis in HEK293A cells, the indicated cells were seeded and treated with IWP-2 overnight. The cells were incubated with anti-V5 antibody diluted in fresh culture medium for 1 hr at 4°C and then washed with PBS to remove unbound antibody. The cells were subsequently treated with control or Wnt5a conditional medium for 1 hr in a humidified incubator at 37°C with 5% $CO_2$. The cells were fixed and permeabilized after treatment, incubated with secondary antibodies, washed with PBS, and stained with DAPI.

## Biotin labeling with miniTurbo

HEK293A ZRDKO cells stably expressing V5-FZD5-mTB or V5-FZD7-mTB, either alone or together with RNF43ΔRING-HA, were treated with IWP-2, control CM, Wnt3a CM, or Wnt5a CM with or without biotin (200 μM) as indicated. Following treatment, the cells were washed with PBS to remove unbound biotin and lysed in lysis buffer (25 mM Tris-HCl (pH 7.4), 150 mM NaCl, 1% NP-40, 1% sodium deoxycholate, 0.6% SDS) containing a protease inhibitor cocktail. The cell lysates were boiled for 10 min at 98°C and centrifuged at 12,000 × $g$ for 15 min. The supernatants were harvested and incubated with NeutrAvidin agarose beads (Thermo Fisher Scientific) overnight. The beads were then centrifuged at 1500 × $g$ for 2 min, washed twice with lysis buffer, once with 1 M KCl, once with 0.1 M $Na_2CO_3$, once with 2 M urea (diluted in 10 mM Tris-HCl, pH 8.0), and twice with lysis buffer. Specific proteins were eluted with 2 × protein loading buffer containing 10 mM biotin and boiled for 10 minutes at 95°C. The eluted protein samples were subsequently analyzed by immunoblotting.

## Statistical analyses

All experiments were repeated at least three times to ensure the reproducibility of the results. For quantification of the number of FZD5 and FZD7 puncta in WT, DVLTKO, and DVLTKO + DVL2 HEK293A cells, images were acquired via a confocal microscope (LSM 980 with Airyscan 2, ZEISS). Punctate structures of FZD proteins in 20 cells per cell line were counted via ImageJ software (NIH Image). To quantify colocalization, Mander's coefficients for FZD5 puncta overlapping with LAMP1 or EEA1 puncta were calculated via ImageJ. Statistical analyses were performed via two-tailed unpaired Student's t tests, and the results are presented as the means ± SDs and were analyzed via GraphPad Prism 8. Statistical significance was defined as follows: ns (no significant difference, p>0.05), *** (p<0.001), **** (p<0.0001).

## Acknowledgements

This research was supported by a grant from the National Natural Science Foundation of China (81870620 to XZ) and by funding from Sichuan Provincial People's Hospital (to XZ) and University of Electronic Science and Technology of China (to XZ). The authors would like to thank the UESTC Public Experiment Platform for Basic Medicine for providing the experimental facilities.

## Additional information

### Funding

| Funder | Grant reference number | Author |
| --- | --- | --- |
| National Natural Science Foundation of China | 81870620 | Xinjun Zhang |
| Sichuan Provincial People's Hospital | | Xinjun Zhang |
| University of Electronic Science and Technology of China | | Xinjun Zhang |

| Funder | Grant reference number | Author |
|---|---|---|

The funders had no role in study design, data collection and interpretation, or the decision to submit the work for publication.

## Author contributions

Dong Luo, Conceptualization, Investigation, Writing – original draft, Writing – review and editing; Jing Zheng, Investigation, Writing – original draft; Shuning Lv, Investigation; Ren Sheng, Maorong Chen, Xi He, Writing – review and editing; Xinjun Zhang, Conceptualization, Supervision, Funding acquisition, Writing – original draft, Writing – review and editing

## Author ORCIDs

Dong Luo (iD) https://orcid.org/0009-0009-4848-4853
Jing Zheng (iD) http://orcid.org/0009-0007-4474-4318
Ren Sheng (iD) https://orcid.org/0000-0002-2293-8986
Maorong Chen (iD) https://orcid.org/0000-0003-3744-8864
Xi He (iD) https://orcid.org/0000-0002-8093-7981
Xinjun Zhang (iD) https://orcid.org/0000-0003-2409-4104

Reviewer #1 (Public review): https://doi.org/10.7554/eLife.103996.3.sa1
Reviewer #2 (Public review): https://doi.org/10.7554/eLife.103996.3.sa2
Author response https://doi.org/10.7554/eLife.103996.3.sa3

# Additional files

## Supplementary files

MDAR checklist

## Data availability

All data generated or analyzed during this study are included in the manuscript and supporting files. Source data files have been provided for all figures.

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
