## [Editor Report · eLife Assessment]

This study presents **important** findings demonstrating that the internalization and degradation of FZD5 and FZD8, two of the ten Frizzled proteins, are WNT dependent and do not involve DVL. The evidence supporting the claims of the authors is **convincing**. This research will be of interest to biologists specializing in Wnt signaling, cancer, and regenerative medicine.

---

## [Referee Report · Reviewer #1 (Public review)]

Summary:

The mechanism by which WNT signals are received and transduced into the cell has been the topic of extensive research. Cell surface levels of the WNT receptors of the FZD family are subject to tight control and it's well established that the transmembrane ubiquitin ligases ZNRF3 and RNF43 target FZDs for degradation and that proteins of the R-spondin family block this effect. This manuscript explores the role that WNT proteins play in receptor internalization, recycling and degradation, and the authors provide evidence that WNTs promote interactions of FZD with the ubiquitin ligases. Using cells mutant in all 3 DVL genes, the authors demonstrate that this effect of WNT on FZD is DVL-independent.

Strengths:

Overall, the data are of good quality and support the authors' hypothesis. Strengths of this study is the use of CRISPR-mutated cell lines to establish genetic requirements for the various components. The finding that FZD internalization and degradation is WNT dependent and does not involve DVL is novel.

Weaknesses:

A weakness of the work includes a heavy reliance on overexpression of FZD proteins. To detect endogenous FZDs, the authors have inserted a V5 tag into the endogenous gene, which may affect their activity(ies).

---

## [Referee Report · Reviewer #2 (Public review)]

In this manuscript Luo et al uncover that the ZNRF3/RNF43 E3 ubiquitin ligases participate in the selective endocytosis and degradation of FZD5/8 receptors in response to Wnt stimulation. In my opinion there are three significant findings of this study: (1) Wnt proteins are required for ZNRF3/RNF43 mediated endocytosis and degradation of FZD receptors and this constitutes an important negative regulatory loop. (2) Wnt can induce FZD endocytosis in the absence of ZNRF3/RNF43 but this does not influence total or cell surface levels. (3) The ZNRF3/RNF43 substrate selectivity for FZD5/8 over the other 8 Frizzleds. Of course, many questions remain, and new ones emerge as it is often the case, but these findings challenge our dogmatic view on how the ZNRF3/RNF43 regulate Wnt signaling and emphasize their role in Wnt-dependent Frizzled endocytosis/degradation and beta-catenin signaling.

This is an elegant study employing several CRISPR-edited cell lines to tag endogenous Frizzled receptors and to knockout ZNRF3/RNF43 and all three Dishevelled proteins. One major strength of the study is therefore the careful assessment of the roles of RNF43 and ZNFR3 in endogenous expression contexts. This is especially relevant since overexpression of membrane E3 ligases have been shown to ectopically degrade membrane proteins and could have blurred previous interpretations. A second strength is clarifying the role of Dishevelled proteins in FZD endocytosis. Indeed, although previous studies suggested that the Wnt-promoted interaction between FZD and RNF43/ZNFR3 was mediated through Dvl, the authors clearly show that this is not the case (using Dvl knockout cells and functional assays). Dvl proteins, on the other han,d are still required for ligand-independent FZD-endocytosis.

The only weakness pertains to the difference in signaling outcome, comparing elevated signaling seen when FZD levels are upregulated following ZNFR3/RNF43 KO vs ectopic overexpression. Indeed, the authors suggest that in the absence of RNF43/ZNFR3 the receptors could be recycled back to the PM and thereby contribute to increased signaling seen in the mutant cells. This has not been directly demonstrated.

---

## [Author Response]

The following is the authors’ response to the original reviews.

**Reviewer #1 (Recommendations for the authors):**
Because many conclusions are drawn from overexpression studies and from a single cell line (HEK293), it is unclear how general these effects are. In particular, one of the main claims put forth in this manuscript is that of specificity, namely, that FZD5/8, and none of the other FZDs, are uniquely involved in this internalization and degradation. While there are examples of similar specificities, many of these examples can be attributed to a particular cellular context. Without demonstrating that this FZD5/8 specificity is observed in multiple cell lines and contexts, this point remains unconvincing and questionable. One way to address this point of criticism is to omit the word "specifically" in the title and soften the language concerning this idea throughout the manuscript.

We appreciate your valuable comments and suggestions. We have removed the word “specifically” from the title and softened the language concerning this idea throughout the manuscript. Moreover, we performed new experiments to show that Wnt3a/5a induces FZD5/8 endocytosis and degradation and that IWP-2 treatment increases the cell surface levels of FZD5/8 in cell lines other than 293A (Figure 1-Figure supplement 1 and Figure 2-Figure supplement 1). These results indicate that Wnt-induced FZD5/8 endocytosis and degradation are not cell specific.

The starting point for these studies is a survey of all 10 FZDs, V5-tagged and overexpressed in HEK293 cells. Here, the authors observed a decline in cell surface levels of only FZD5 and 8 in response to Wnt3a and Wnt5a. As illustrated in the immunoblot (Fig 1B), several FZDs were poorly expressed, including FZD1, 3, 6 and 9, which calls into question that only FZD5 and 8 were affected. Furthermore, total levels of FZD8 don't diminish appreciably, as claimed by the authors, and only FZD5 shows a subtle decline upon WNT treatment. All of these experiments are performed with overexpressed V5-tagged FZD proteins or with endogenously V5-tagged (KI) proteins, and it is possible that overexpression or tagging lead to potentially artifactual observations. Examining the effects of WNTs on FZD protein localization and levels need to be done with endogenously expressed, non-tagged FZDs. In this context, it is somewhat puzzling that the authors don't show such an experiment using the pan- and FZD5/8-specific antibodies, which they use in multiple experiments throughout the manuscript. With these available tools it should be possible to examine FZD levels at the cell surface in response to Wnt3a and Wnt5a, ideally in multiple cell lines.

We appreciate your valuable comments and suggestions. Figure 1B shows the results of the follow-up study shown in Figure 1A. As shown in Figure 1A, we used flow cytometry analysis to detect the cell surface levels of stably expressed FZDs and found that Wnt3a/5a specifically reduced the levels of FZD5/8 on the cell surface, suggesting that Wnt3a/5a induces FZD5/8 endocytosis. As shown in Figure 1B and C, we performed immunoblotting to examine whether Wnt3a/5a-induced FZD5/8 internalization resulted in FZD5/8 degradation. Notably, most FZDs exhibit two bands on immunoblots, as also suggested by other published studies, and the upper bands represent the mature form that is fully glycosylated and presented to the cell surface (see also new Figure 2L), whereas the lower bands represent the immature form. Our results clearly indicated that Wnt3a/5a treatment reduced the levels of the mature forms of both FZD5 and FZD8, although the immunoblotting signals of the mature form of FZD8 (upper bands) were relatively weak. The immunoblotting signals of the other FZDs varied, and some of them (including FZD1, -3, -6 and -9) were relatively weak; however, according to the results in Figure 1A, all of the FZDs were expressed and present on the cell surface.

Commercially available FZD5/8 antibodies, including those used in published studies, cannot detect endogenous FZD5/8 or can only recognize immature FZD5 in our hands, which is why we have to use the CRISPR-CAS9-based KI technique to introduce a V5 tag to FZD5 and FZD7. Notably, in the overexpression experiments, the V5 tag is on the amino terminus, and in the KI experiments, the V5 tag is on the carboxyl terminus of FZDs, which may minimize the potential artificial effects of the V5 tag on the immunoblotting assays.

The monoclonal antibodies used in this study, such as anti-pan-FZD, anti-FZD5/8, and anti-FZD4 antibodies, are neutralizing antibodies that can compete with Wnt ligands to bind to the FZD CRD. These antibodies have been successfully used to detect the surface levels of FZDs via flow cytometry assays. However, as the binding affinity of the Wnt-FZD CRD is comparable to the binding affinity of the antibody-FZD, we were cautious in using these antibodies to detect the cell surface levels of FZDs when the cells were treated with Wnt3a/5a CM, which contains relatively high concentrations of Wnt3a/5a. As shown in Author response image 1, Wnt3a or Wnt5a treatment dramatically reduced the endogenous cell surface level of FZD5/8, as detected by flow cytometry using the anti-FZD5/8 antibody. However, in another experiment, HEK293A cells were first incubated with cold Wnt3a or Wnt5a CM at 4°C to minimize endocytosis and then analyzed via flow cytometry using the anti-FZD5/8 antibody. The results showed that Wnt3a/5a incubation reduced the floe cytometry signals, suggesting that Wnt3a/5a binding to FZD5/8 might interfere with antibody-FZD5/8 binding, although we cannot exclude the possibility that Wnt3a/5a may induce FZD5/8 endocytosis at 4°C (Author response image 1).Author response image 1

**Author response image 1. sa3fig1:** High concentrations of Wnt3a or Wnt5a may interfere with the recognition of FZD5/8 by the anti-FZD5/8 antibody in flow cytometry assays. (A) HEK293A cells were treated with control, Wnt3a or Wnt5a CM for 2 hours at 37°C in a humidified incubator and were analyzed via flow cytometry using the anti-FZD5/8 antibody. (B) HEK293A cells were incubated with control, Wnt3a or Wnt5a CM for 1 h at 4°C and analyzed by flow cytometry using the anti-FZD5/8 antibody.

Several experiments rely on gene-edited clonal cell lines, including knockouts of FZD5/8, RNF43/ZNRF3, and DVL. Gene knockouts were confirmed by genomic DNA sequencing and, for DVL and FZD5/8, by loss of protein expression. While these KO lines are powerful tools to study gene function, there is a concern for clonal variability. Each cell line may have acquired additional changes as a result of gene editing. In addition, there may be compensatory changes in gene expression as a consequence of the loss of certain genes. For example, expression of other FZDs may increase in FZD5/8 DKO cells. To address this critique, the authors should show that re-expression of the knocked-out genes rescues the observed effect. This is done in some instances (Fig 5E, G, H) but not in other instances, such as with the DVL TKO (Fig. 3). Since the authors assert that DVL is important for FZD internalization in the absence of WNT, but not for FZD internalization in the presence of WNT, this particular rescue experiment is important. This is a potentially important finding and it should be confirmed by re-expression of DVL in the TKO line. As an alternative, conditional knockdown using Tet-inducible shRNA expression could address concerns for clonal variability.

We appreciate your valuable comments and suggestions. We re-expressed DVL2 in DVLTKO cells stably expressing V5-linker-FZD5 or V5-linker-FZD7. As shown in Figure 3G-K, re-expression of DVL2 rescued the decreased Wnt-independent endocytosis of FZD5 and FZD7 caused by DVL1/2/3 knockout.

Given the significant differences in signaling activity by Wnt3a and Wnt5a, it is somewhat surprising that all experiments shown in this manuscript do not identify distinguishing features between Wnt3a and Wnt5a. In addition, it is unclear why the authors switch between Wnt3a and Wnt5a. For example, Figures 1C, 3G-J, 4C-D only use Wnt5a. In contrast, Figures 6E and H use Wnt3a, most likely because b-catenin stabilization is examined, an effect generally not observed with Wnt5a. The choice of which Wnt is examined/used appears to be somewhat arbitrary and the authors never provide any explanations for these choices. In the end, this type of inconsistency becomes puzzling when the authors present, quite convincingly, in Figure 7, that both Wnt3a and 5a promote an interaction between FZD5/8 and RNF43 through proximity biotin labeling.

Although Wnt3a and Wnt5a are significantly different in triggering intracellular signaling pathways, both bind FZD5/8 and induce FZD5/8 endocytosis and degradation similarly. When FZD5 is stably overexpressed, Wnt5a has slightly stronger effects on inducing FZD5 endocytosis and degradation, possibly because the Wnt5a concentration may be higher than the Wnt3a concentration in our CM, which is why we used Wnt5a CM in some experiments when V5-FZD5 was overexpressed. In the revised manuscript, we used both Wnt3a and Wnt5a CM in the experiments as you suggested, as shown in Figure 1C, 3G-K and Figure 4-Figure supplement 1.

Minor Points:Figure 3G and I: it is curious that individual cells are shown in the "0 h" samples, while the "Con 1 h" and "Wnt5a 1 h" show multiple cells with several making direct contact with each other. This is notable because the V5 staining at sites of cell-cell contact are quite distinct and variable between control and Wnt5a-treated and WT versus DVL TKO cells. Also, sub-cellular localization of FZD5 (V5 tag) puncta is quite distinct between Con and Wnt5a: puncta in Wnt5a-treated cells appear to be more plasma membrane proximal than in Con cells. These points may be easy to address by showing images of cells that are more similar with respect to cell number and density for each condition.

Thank you for your suggestions. We repeated these experiments and added Wnt3a treatment and adjusted the cell density. Images including an individual cell were selected for presentation.

Figure 5E: the following statement is confusing/misleading: "Furthermore, reintroducing ZNRF3 or RNF43 into ZRDKO cells efficiently restored the increase in cytosolic β-catenin levels, whereas the expression of RNF130 or RNF150, two structurally similar transmembrane E3 ubiquitin ligases, did not (Fig. 5E)." First, reintroduction of ZNRF3 or RNF43 restores cytosolic b-catenin levels; it does not restore the increase in b-catenin. Second, the claim that RNF130 fails to have this effect is not substantiated since it is barely expressed.

Thank you for your suggestions and comments. We reorganized the language to make the statement clearer. Notably, the expression level of RNF130 was relatively low compared with that of other E3 ligases, but RNF130 was expressed (Figure 5E darker exposure) and could reduce the cell surface levels of FZDs, as shown in Figure 5G.

**Reviewer #2 (Recommendations for the authors):**
(1) Given their results the authors conclude that upregulation of Frizzled on the plasma membrane is not sufficient to explain the stabilization of beta-catenin seen in the ZNRF3/RNF43 mutant cells. This interpretation is sound, and they suggest in the discussion that ZNRF3/RNF43-mediated ubiquitination could serve as a sorting signal to sort endocytosed FZD to lysosomes for degradation and that absence or inhibition of this process would promote FZD recycling. This should be relatively easy to test using surface biotinylation experiments and would considerably strengthen the manuscript.

Thank you for your valuable suggestions and comments. We performed cell surface biotinylation experiments in HEK293A FZD5KI cells, as shown in Figure 2L. The results indicated that Wnt3a or Wnt5a treatment induced the degradation of FZD5 on the cell surface, which was antagonized by cotreatment with RSPO1. We did not perform a more detailed endocytosis/recycling biotinylation experiment that requires complex reversible biotinylation and multiple washing steps because HEK293A cells are fragile in culture and not easy to handle. Furthermore, the results shown in Figure 4 indicate that knockout of ZNRF3/RNF43 or RSPO1 significantly blocked the degradation of internalized FZD5 and reduced the colocalization of internalized FZD5 with lysosomal markers, suggesting that Wnt3a/5a induced lysosomal degradation of FZD5 in the presence of ZNRF3/RNF43 and that the internalized FZD5 was most likely recycled back to the cell surface when ZNRF3/RNF43 was knocked out or inhibited by RSPO1.

(2) The authors show that the FZD5 CRD domain is required for endocytosis since a mutant FZD5 protein in which the CRD is removed does not undergo endocytosis. This is perhaps not surprising since this is the site of Wnt binding, but the authors show that a chimeric FZD5CRD-FZD4 receptor can confer Wnt-dependent endocytosis to an otherwise endocytosis incompetent FZD4 protein. Since the linker region between the CRD and the first TM differs between FZD5 and FZD4, it would be interesting to understand whether the CRD specifically or the overall arrangement (such as the spacing) is the most important determinant.

Our results in Figure 1D-H clearly show that the CRD of FZD5 specifically is both necessary and sufficient for Wnt3a/5a-induced FZD5 endocytosis, as replacing the CRD alone in FZD5 with the CRD from either FZD4 or FZD7 completely abolished Wnt-induced endocytosis, whereas replacing the CRD alone in FZD4 or FZD7 with the FZD5 CRD alone could confer Wnt-induced endocytosis.

(3) I find it surprising that only FZD5 and FZD8 appear to undergo endocytosis or be stabilized at the cell surface upon ZNRF3/RNF43 knockout. Is this consistent with previous literature? Is that a cell-specific feature? These findings should be tested in a different cell line, with possibly different relative levels of ZNRF3 and RNF43 expression.

Thank you for your comments and suggestions. Our finding that ZNRF3/RNF43 specifically regulates FZD5/8 degradation is consistent with recent published studies in which FZD5 is required for the survival of RNF43-mutant PDAC or colorectal cancer cells (Nature Medicine, 2017, PMID: 27869803) and FZD5 is required for the maintenance of intestinal stem cells (Developmental Cell, 2024, PMID: 39579768 and 39579769), and in both cases, FZDs other than FZD5/8 are also expressed but not sufficient to compensate for the function of FZD5. The mechanism by which Wnt3a/5a specifically induces FZD5/8 endocytosis and degradation is currently unknown and needs to be explored in the future. We speculate that Wnt binding to FZD5/8 may recruit another protein on the cell surface to specifically facilitate FZD5/8 endocytosis. On the other hand, we cannot exclude the possibility that Wnts other than Wnt3a/5a may induce the endocytosis and degradation of FZDs other than FZD5/8 since there are 19 Wnts and 10 FZDs in humans. Notably, several previous studies have suggested that ZNRF3/RNF43 may regulate the endocytosis and degradation of all FZDs without selectivity (such as Nature, 2012, PMID: 22575959; Nature, 2012, PMID: 22895187; Mol Cell, 2015, PMID: 25891077). However, their conclusions were drawn mostly on the basis of overexpression studies. According to the results shown in Figure 5E-H, overexpressing a membrane-tethered E3 ligase (such as ZNRF3, RNF43, RNF130, or RNF150) may nonspecifically degrade FZD proteins on the cell surface.

Furthermore, in the revised manuscript, we showed that Wnt3a/5a induced FZD5/8 endocytosis and degradation in multiple cell lines, including Huh7, U2OS, MCF7, and 769P cells (Figure 1-Figure supplement 1 and Figure 2-Figure supplement 1), suggesting that these phenomena are not specific to 293A cells.

(4) If FZD7 is not a substrate of ZNRF3/RNF43 and therefore is not ubiquitinated and degraded, how do the authors reconcile that its overexpression does not lead to elevated cytosolic beta-catenin levels in Figure 5B?

We are currently not sure of the mechanism underlying this result. Considering that most FZDs are expressed in 293A cells, we do not know how much of the mature form of overexpressed FZD7 was presented to the plasma membrane.

(5) For Figure 5B, it would be interesting if the authors could evaluate whether overexpression of FZD5 in the ZNRF3/RNF43 double knockout lines would synergize and lead to further increase in cytosolic beta-catenin levels. As control if the substrate selectivity is clear FZD7 overexpression in that line should not do anything.

Thank you for your suggestion. We performed these experiments as suggested, and the results indicated that overexpressing FZD5 further increased cytosolic beta-catenin levels in ZRDKO cells, whereas FZD7 had no effect (Figure 6D).

(6) In Figure 6G, the authors need to show cytosolic levels of beta-catenin in the absence of Wnt in all cases.

We did not add Wnt CM in this experiment. RSPO1 activity, which relies on endogenous Wnt, has been well documented in previous studies.

(7) Since the authors show that DVL is not involved in the Wnt and ZRNF3-dependent endocytosis they should repeat the proximity biotinylation experiment in figure 7 in the DVL triple KO cells. This is an important experiment since previous studies showed that DVL was required for the ZRNF3/RNF43-mediated ubiqtuonation of FZD.

Thank you for your valuable suggestions. As you suggested, we performed a proximity biotinylation experiment in DVL TKO cells, and the results showed that Wnt3a/5a could still induce the interaction of FZD5 and RNF43 in DVLTKO cells (Figure 7-figure supplement 1), suggesting that the Wnt-induced FZD5‒RNF43 interaction is DVL independent.